



# Methods to characterize type, relevance, and interactions of organic matter and microorganisms in fluids along the flow path of a geothermal facility

Alessio Leins[1,2], Danaé Bregnard[3], Andrea Vieth-Hillebrand[1], Stefanie Poetz[1], Florian Eichinger[4], Guillaume Cailleau[3], Pilar Junier[3], and Simona Regenspurg[1]

[1]GFZ German Research Centre for Geosciences, Telegrafenberg, 14473, Potsdam, Germany
[2]Friedrich Schiller University, Institute of Geosciences, Applied Geology, Burgweg 11, 07749 Jena, Germany
[3]University of Neuchâtel, Laboratory of Microbiology, Institute of Biology, Rue Emile-Argand 11, 2000 Neuchâtel, Switzerland
[4]Hydroisotop GmbH, Woelkestraße 9, 85301 Schweitenkirchen, Germany

**Correspondence:** Alessio Leins (leins@gfz-potsdam.de)

**Abstract.** Dissolved organic matter and microorganisms were analyzed along the flow path of a geothermal facility in Austria. Various analytical methods were used to characterize and differentiate between natural and synthetic organic matter, characterize the microbial community composition, and determine the implications of microorganisms in an operating a geothermal site. Dissolved organic carbon (DOC) concentrations were in the range of 8.4–10.3 $\mathrm{mg\,C\,L^{-1}}$ and typically decreased from the production to the injection side. Carbonate scalings are avoided in the facility by the injection of a chemical scaling inhibitor within the production well at 500 m depth. It was calculated that the inhibitor contributes approximately 1 $\mathrm{mg\,C\,L^{-1}}$ DOC to the produced fluids. Ion chromatography (IC), liquid chromatography — organic carbon detection (LC-OCD) and Fourier-transform ion cyclotron resonance mass spectrometry (FT-ICR-MS) in negative electrospray ionization (ESI(-)) and positive atmospheric pressure photoionization (APPI(+)) mode were applied to the fluid samples to characterize the dissolved organic matter (DOM) composition and distinguish between the inhibitor and the natural DOM. Depending on the applied ionization mode, FT-ICR-MS results show that between 31 % and 65 % of the macromolecular formulas detected in the fluid samples seem to originate from the inhibitor. However, the DOM is mainly composed of low molecular weight acids (LMWA), especially acetate with up to 7.4 $\mathrm{mg\,C\,L^{-1}}$. The microbial community composition varied along the flowpath with dominant phyla being Firmicutes, Proteobacteria, and Thermotogae. Based on the microorganisms found in the sample, the metabolic pathways have been assessed. Acetate might be produced by microorganisms through various fermentation processes (e.g. from lysine, pyruvate and hexitol). Assessing the presence and interaction of organic compounds and microorganisms in geothermal fluids provides a broader understanding of processes within the geothermal facility. This understanding could be beneficial for the efficient use of a geothermal power plant.



## 1 Introduction

Deep hydrothermal energy production is increasingly gaining in importance as an alternative energy source. Geothermal power plants extract the heat of subsurface fluids to produce heat and electricity. The depths at which these fluids are extracted may vary from a few hundred meters to a few kilometers. Geothermal power plants encounter many operating challenges such as mineral precipitation (scaling) or corrosion of the casing of the boreholes and geothermal plant components (Regenspurg et al., 2016; Demir et al., 2014). Scaling and corrosion are caused by hydrochemical reactions linked to pressure and temperature

changes of the fluid during transport or to the presence of metabolic byproducts of microorganisms present in the fluids (Inagaki et al., 2003; Little and Lee, 2015). Therefore, it is crucial to analyze the fluids and their composition to better understand, predict, and mitigate possible chemical reactions that might compromise the functioning of the facilities of a geothermal plant. However, to date, the role of organic components is rarely considered as part of the analyses performed in the characterization of geothermal power stations. Moreover, only in recent studies corrosion and declining injectivity due to the formation of

biofilms, microbially induced corrosion (MIC) or microbially-induced mineral scaling in power plants have been reported (Alawi et al., 2011; Vetter, 2012; Lerm et al., 2013; Little and Lee, 2015; Westphal et al., 2019; Brehme et al., 2020; Leins et al., 2022; Madirisha et al., 2022). The growth of those microorganisms may be supported by organic compounds in the fluids. These organic compounds derive either from natural sources within the geothermal aquifer or might have been added artificially, for instance from detergents during drilling or by the injection of organic scaling inhibitors.

Organic acid anions were reported to be present in a variety of deep subsurface systems such as oil-field waters (Carothers and Kharaka, 1978; Hatton and Hanor, 1984; Kharaka et al., 1985, 1997), waters from fractured crystalline rock (Sherwood Lollar et al., 2021; Kieft et al., 2018), hydrothermal vents (Lang et al., 2010, 2018; McDermott et al., 2015), and fluids from geothermal sites of the Molasse Basin (Alawi et al., 2011; Vetter, 2012; Leins et al., 2022). Their concentrations in the fluids can vary by several orders of magnitude ranging from a few $mg\,L^{-1}$ to 10,000 $mg\,L^{-1}$ (Kharaka et al., 1997). The highest

concentrations are typically found in oil-field waters and are dominated by acetate, followed by propionate, butyrate, and valerate (Carothers and Kharaka, 1978; Fisher and Boles, 1990; Kharaka et al., 1987).

For information on the presence of specific organic acid anions in the fluids and the scaling inhibitor, IC was applied. These organic acid anions however, might form only a small fraction of the detectable DOC present in the fluids. In this study, DOC characterization was conducted via LC-OCD to not only quantify the DOC content, but also characterize its distribution into

fractions with size exclusion chromatography (SEC). These fractions vary from low molecular weight compounds, such as low molecular weight acid (LMWA) and neutral (LMWN) compounds, to high molecular weight compounds (e.g. humic substances and biopolymers). These analyses were conducted on fluids and a scaling inhibitor sample from the geothermal power plant of Bad Blumau, Austria. Furthermore, a more detailed characterization of medium to high molecular weight organic compounds within a mass range from 150 to 1000 Da was carried out with Fourier transform ion cyclotron resonance mass spectrometry

(FT-ICR-MS) in both APPI(+) and ESI(-). FT-ICR-MS enables the determination of elemental formulas by providing accurate masses of the molecules. This allows to reveal the influence of chemical scaling inhibitors and biomarkers for the presence of microorganisms in the geothermal fluids on a molecular level. To the best of our knowledge, this is the first time that FT-ICR-



MS was used for the characterization of DOM in fluids from a geothermal power plant. FT-ICR-MS has already been applied to a variety of water systems such as groundwater (McDonough et al., 2020), deep fracture water (Kieft et al., 2018), pore water (D'Andrilli et al., 2010; Rossel et al., 2016; Schmidt et al., 2009), hydrothermal vents (Noowong et al., 2021; Rossel et al., 2017; Gomez-Saez et al., 2016), and marine as well as terrestrial waters (D'Andrilli et al., 2010; Koch et al., 2008; Minor et al., 2012; Sleighter and Hatcher, 2008).

In addition to our DOM analyses, the analysis of the bacterial diversity was conducted in parallel in the fluids. In the present study, targeted amplification of the 16S rRNA gene allowed to assess the bacterial diversity present at the three same sampling points. Furthermore, bacterial metabolic pathways were predicted based on the known metabolisms of the most dominant microorganisms found in our samples, to explain the presence of certain organic compounds. The bacterial metabolic pathways linked to the consumption or production of acetate were accessed in order to see if changes in the microbial community are linked with changes in the presence of specific organic acids. Changes in the bacterial community, the associated changes of the metabolic pathways present, as well as microbial growth itself may impact the efficacy of power plants, underlining the importance to asses both the organic compounds and microbial composition of geothermal fluids.

This study aims to (1) characterize the DOM and microbial community of a deep geothermal fluid by various methods; (2) to distinguish with these methods between natural and synthetic DOM and determine its origin (3) to determine the metabolic pathways linked to acetate consumption or production, and (4) to asses if the DOM composition correlates with a change in microbial diversity.

## 2 Material and methods

### 2.1 Site description

The geothermal site Bad Blumau is a geothermal power plant and thermal spa located in south-east Austria (Upper Styrian Basin as part of the Pannonian Basin). The targeted geothermal system is also used by several other spas and heat usages in the area, and was discovered during a hydrocarbon exploration campaign in the second half of the 20th century (Alt-Epping et al., 2013). The Styrian Basin is of Miocene age, composed of Tertiary siliciclastic basin-fills underlain by Paleozoic carbonates and phyllites of the Grazer Paleozoicum, which overlie the crystalline basement (Goldbrunner, 2000; Alt-Epping et al., 2013). The geothermal reservoir in this area is hosted by carbonate rocks, which consist of Devonian limestones and dolomites originating from Paleozoic reef development (Hubmann et al., 2006). Tectonic deformation caused intense fracturing of the carbonate rocks, which therefore exhibit good aquifer properties (Goldbrunner, 2000).

In addition to its balneological purpose, the site is also used for heat and electricity production, as well as the commercial production of liquefied $CO_2$ (Alt-Epping et al., 2013). The system is operated as a geothermal doublet with an internal distance of 2300 m between the production and injection well and targets the Paleozoic (Devonian) carbonate formation in a depth of 2800 m (Goldbrunner, 2000). The reservoir temperature was reported to be 124 °C (Goldbrunner, 2005), while the produced fluids at the production well head reach 107 °C and are reinjected with approximately 65–50 °C after the heat extraction (Westphal et al., 2019). The geothermal fluid ascends via natural gas lift of $CO_2$ with an average flow rate of 20 L s$^{-1}$. The $CO_2$




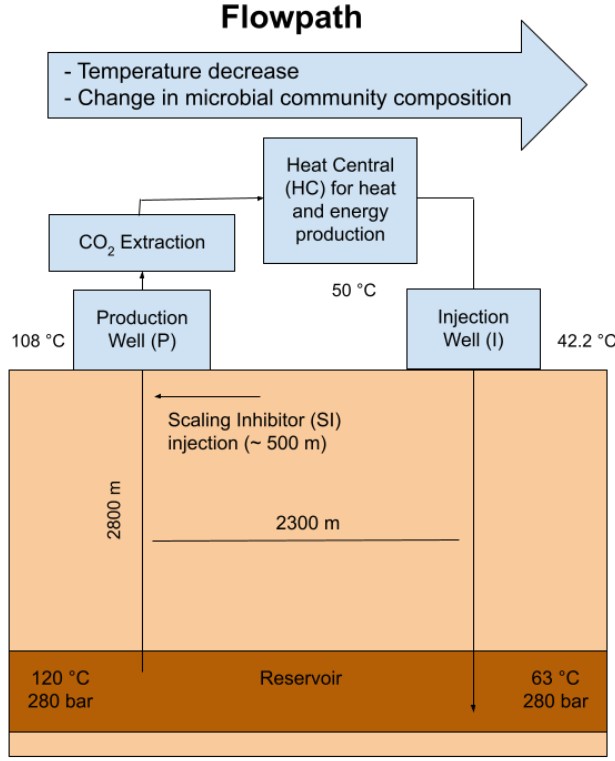

**Figure 1.** Schematic of the geothermal power plant in Bad Blumau. The black dots indicate the fluid sampling points for the production well (P), heat central (HC), and injection well (I). Modified from Westphal et al. (2019).

is reported to ascend from the mantle regions along fault zones and is regarded to be a product of the Neogene volcanism of the Styrian Basin (Goldbrunner, 2000). The $CO_2$ in the system poses a major challenge for the plant operation. At approximately 300–350 m below the surface, the exsolution of the $CO_2$ takes place leading to carbonate precipitation (Alt-Epping et al., 2013).

To prevent carbonate scaling, the inhibitor named hydrin 45.3 is injected into the fluid within the production well at a depth of approximately 500 m, at a pressure before exsolution of $CO_2$. The current inhibitor consists of organic polyelecrolytes. Generally, the exact chemical composition of the inhibitor falls under the protection of commercial and industrial secrecy.

    The geothermal fluids at Bad Blumau were described to be of $NaHCO_3$ type with a salinity of approximately 20 g L$^{-1}$, and a slightly alkaline pH of around 8.0 (Westphal et al., 2019). The $CO_2$ concentration makes up 99 % of the gases and is estimated 95   to be approximately 5 L L$^{-1}$.



## 2.2 Sample collection

Fluid samples were collected during two sampling campaigns in March and June 2021. In both campaigns one sample was taken from each sampling point from the surface installation during regular operation of the geothermal plant, at the production well GB2 (P) before $CO_2$ extraction, at the heat central (HC), and at the injection well GB1 (I) (Fig. 1). The respective fluid

where at approximately 108 °C (P), 50 °C (HC), and 42.2 °C (I). Unfiltered fluids were collected in 500 ml Duran glass bottles with screw caps containing teflon-coated septa inside. The bottles were pre-rinsed with the fluid and afterwards filled completely to avoid contact with air. The samples were stored at 4 °C until shipment to the laboratory, where they were again stored at 4 °C until further analyses. In addition to the fluid samples, a sample of the used scaling inhibitor was obtained from the site operators.

For microbial analysis, 40 L of fluids were sampled at the same three sampling points (P, HC, I) during the sampling campaign in June 2021. The 40 L of fluids were directly filtered through 0.22 μm nitrocellulose membrane filters (Merck Millipore, Germany) under sterile conditions. Six independent filters were prepared simultaneously as independent replicates using an EZ-Stream pump (Merck Millipore, Germany) and six glass filtration stations mounted on a manifold (Merck Millipore, Germany). Filters were transported at 4 °C to the laboratory, where they were stored at -20 °C until further processing.

## 110 2.3 Analytical Methods

### 2.3.1 Ion chromatography

The quantification of organic anions (formate, acetate, propionate, butyrate, valerate, oxalate) and inorganic anions ($F^-$, $Cl^-$, $Br^-$, $SO_4^{2-}$) from both sampling campaigns was conducted via IC (ICS 3000, Thermo Fisher Scientific) using an AS-AP autosampler, AS11 HC column and a conductivity detector. KOH solutions with varying concentrations over time were used

as eluent for the samples. The initial KOH concentration was 1.4 mM and stepwise increased towards 60 mM within 32 min. After 32 min the concentration was reduced to the initial value of 1.4 mM and equilibrated for 12 min. The flow-rate was 0.38 ml min$^{-1}$. The column temperature was at 35 °C and 10 μl of sample was injected for each run. The quality of the measurements was verified daily using standards that contain the analytes in different concentrations. The concentrations were 0.02; 1.0; 10 and 100 mg L$^{-1}$. For samples with high chloride concentrations (>1 g L$^{-1}$), the chloride was reduced prior to the

analysis of the organic anions using OnGuard II AG/H cartridges (Thermo Fischer Scientific).

### 2.3.2 Liquid chromatography – organic carbon detection (LC-OCD)

The characterization and quantification of the DOC and its fractions from both sampling campaigns were determined by SEC with subsequent UV ($\lambda$ = 254 nm) and IR detection by a LC-OCD system (Huber and Frimmel, 1996). Phosphate buffer (pH 6.85; 2.7 g L$^{-1}$ KH$_2$PO$_4$, 1.6 g L$^{-1}$ Na$_2$HPO$_4$) was used as mobile phase and a flow of 1.1 mL min$^{-1}$ was adjusted (Huber et al.,

2011). The samples passed a 0.45 μm membrane syringe filter before entering the chromatographic column (Toyopearls HW 50 S, 30 μm 250 mm x 20 mm). Here, the DOC will be separated into different fractions according to their molecular masses:





**Table 1.** Description of LC-OCD fractions (Zhu et al., 2015). Modified from Huber et al. (2011), Penru et al. (2013).

| Fraction | Molecular Mass Range | Properties | Description |
| --- | --- | --- | --- |
| Hydrophobic organic carbon (HOC) | | Hydrophobic | lipids (fats) released from bacteria and algae, hydrocarbons |
| Biopolymers (Makro.1) | >10,000 Da | Not UV-absorbable, hydrophilic | Polysaccharides and proteins |
| Humic substances (Makro.2) | ∼1000 Da | Highly UV-absorbable, hydrophobic | Calibration based on Suwannee River standard from IHSS |
| Building blocks (Makro.3) | 350–500 Da | UV-absorbable | Breakdown products of humic substances |
| Low molecular weight organic acids (LMWA) | <350 Da | Negatively charged | aliphatic acids |
| Low molecular weight neutrals (LMWN) | <350 Da | Weakly or uncharged hydrophilic, amphiphilic | Alcohols, aldehydes, ketones, amino acids |

Macro.1 (biopolymers), Macro.2 (humic substances), Macro.3 (building blocks), low molecular weight acids (LMWA), and low molecular weight neutrals (LMWN) (Huber et al., 2011). See Table 1 for properties and description of the fractions. The DOC was quantified by IR detection of the released $CO_2$ after UV oxidation ($\lambda = 185\,nm$) in a Gräntzel thin-film reactor. Humic and

fulvic acids standards of the Suwannee River, provided by the International Humic Substances Society (IHSS), were used for molecular mass calibration. Solutions of known amounts of potassium hydrogen phthalate were used for external calibration of the $CO_2$-quantification.

### 2.3.3 Solid phase extraction (SPE)

Salts are known to cause ionization suppression (King et al., 2000) and have to be eliminated prior to FT-ICR-MS analysis. On

the other hand, also the concentrations of DOM in natural geothermal water samples is too low for being analyzed directly by FT-ICR-MS. Therefore, geothermal fluids had to be pretreated by solid phase extraction (SPE) on SPE cartridges (PPL Bond Elut 1 g, 6 ml cartridge; Agilent Technologies, Germany) (Dittmar et al., 2008) to obtain salt free samples and accumulate 1 mg of DOC for the FT-ICR-MS analysis. The cartridges were pre-rinsed with methanol and acidified deionized water (pH2, hydrochloric acid) for cleaning. The samples were filtered with 0.45 µm membrane syringe filters, diluted with deionized water

(1:1) and acidified up to pH 2 with hydrochloric acid (suprapur) and passed through the cartridges. The sample amount was adjusted to approximately contain 1 mg of DOC. After the absorption, the cartridges were rinsed with 3 x 6 ml of acidified deionized water (pH 2) to remove any remaining salts. The cartridges were dried by vacuum pump for 5 minutes. Finally, the DOM was eluted with 6 ml methanol into pre-combusted glass vials, dried under $N_2$ atmosphere and weighed. The dried samples were then stored in the dark at -24 °C until FT-ICR-MS analysis.





### 2.3.4 Fourier transform ion cyclotron resonance mass spectrometry (FT-ICR-MS)

FT-ICR-MS with its ultra-high resolution in combination with atmospheric pressure ionization modes can provide the elemental composition of thousands of individual medium- to high-molecular weight organic compounds. All the DOM samples as well as the inhibitor were dissolved in methanol (MeOH) to give a stock solution with a final concentration of 1 $mg\,ml^{-1}$. The samples were analyzed on a Bruker Solarix FT-ICR-MS with a 12 T refrigerated actively shielded superconducting magnet. For ESI(-) analysis, the stock solutions were spiked with 4 μL of 25 % aqueous ammonia solution. Measurement solutions of 100 $\mu g\,ml^{-1}$ in MeOH were prepared. Ionization was realized with an Apollo II ESI source from Bruker Daltonik GmbH (Bremen, Germany) in negative ion mode. Samples were infused at a flow rate of 150 $\mu l\,h^{-1}$ using a syringe pump (Hamilton). The capillary voltage was set to 3000 V and an additional collision-induced dissociation (CID) voltage of 70 V in the source was applied to avoid cluster and adduct formation. Nitrogen was used as drying gas at a flow rate of 4.0 $L\,min^{-1}$ and a temperature of 220 °C and nebulizing gas at 1.4 bar. The spectra were recorded in broadband mode using 4 megaword data sets. Ion accumulation time was set to 0.05 s and 200 scans were collected and added to each mass spectrum. Ions were detected in a *m/z* range between 150 and 1000.

For the APPI(+) analyses, measurement solutions of 20 $\mu g\,ml^{-1}$ in MeOH were prepared from the stock solutions. The ion source was a APPI-II from Bruker Daltonik GmbH (Bremen, Germany). Samples were introduced into the MS at an infusion flow rate of 20 $\mu l\,min^{-1}$ with a syringe pump (Hamilton). The capillary voltage was set to -1000 V and CID to 30 V. Nitrogen was used as drying gas at a flow rate of 3.0 $L\,min^{-1}$ and temperature of 210 °C as well as nebulizing gas at 2.3 bar and temperature of 350 °C. The spectra were recorded in broadband mode using 4 megaword data sets. Ion accumulation time was set to 0.05 s and 300 scans were collected and added to each mass spectrum. Ions were detected in a *m/z* range between 147 and 1500.

In ESI(-), the DOM samples were internally recalibrated using $O_x$ compounds, while for the inhibitor both $O_x$ and $S_1O_x$ compounds were used. In APPI(+), both the fluid samples and the inhibitor sample were internally recalibrated using $O_x$ compounds. The root mean square deviations of the eight internal calibrations ranged between 0.013 and 0.018. Method blanks covering sample preparation steps (SPE) and the FT-ICR-MS measurement were prepared for both modes and blank signals were removed from the DOM signal list of the fluid and the inhibitor samples.

Data evaluation was done with the software packages Data Analysis 4.0 SP5 (Bruker Daltonik GmbH, Germany), Excel 2019 (Microsoft Corporation, Redmont, WA), and the statistical data analysis tool R 4.0.1 (R Core Team, 2020) using the tidyverse package (Wickham et al., 2019). Only *m/z* values with a signal to noise ratio $\geq$ 9 were exported for formula assignment. The molecular formulas were calculated by considering $^{12}C$ and $^{13}C$ isotopes with upper elemental thresholds of O $\leq$ 32, S = 1, and N = 1, C and H were unlimited. The mass tolerance was set to ± 0.5 ppm and formulas containing $^{13}C$ were excluded from the final dataset.

The double bond equivalent (DBE) is a measure to express the number of double bonds and rings. With the respective molecular formula it can be calculated from the number of the carbon (C), hydrogen (H), and nitrogen (N) atoms as follows in Eq. (1):



$$DBE = C - \frac{H}{2} + \frac{N}{2} + 1 \qquad (1)$$

Since the DBE counts all double bonds with at least one carbon as a bonding partner, it is not well suited to describe aromaticity of DOM compounds that contain a high number of double bonds within carboxy groups. Therefore the DOM adapted modified aromaticity index ($AI_{mod}$) described by Koch and Dittmar (2006) has been used as expressed in Eq. (2) to evaluate the proportion of aromatic compounds in the dataset. It is a measure for the double bond density in a molecule by considering the contribution of heteroatoms. The $AI_{mod}$ is based on the assumption that 50 % of the oxygen is bound with
double bonds in carboxyl groups.

$$AI_{mod} = \frac{1 + C - 0.5O - S - 0.5(N + P + H)}{C - 0.5O - N - S - P} \qquad (2)$$

Three ranges were established to describe the aromaticity of a given DOM compound. $AI_{mod}$ values $\leq 0.5$ are described as aliphatic, $AI_{mod}$ between 0.5 and 0.67 represent aromatic compounds, and $AI_{mod} \geq 0.67$ describes condensed aromatic compounds (Koch and Dittmar, 2006)

Intensity-weighted averages for DBE, $AI_{mod}$, O/C ratio, H/C ratio, carbon, hydrogen, and oxygen number in each sample were calculated after Bae et al. (2011) in Eq. (3)

$$var_{average} = \frac{\Sigma_i Ii * (var)i}{\Sigma_i Ii} \qquad (3)$$

where *Ii* and *(var)i* are the relative abundances and respective variable value of peak *i*.

### 2.3.5 Microbial analysis

Five of the six filters were processed for DNA extraction. The last filter was kept as a backup. DNA extraction was done with the FastDNA®SPIN kit for soil (MP Biomedicals, USA) using three bead-beating rounds and pooling the three independent DNA extracts at the final step (Wunderlin et al., 2013). In parallel a DNA blank extract was prepared by performing the same procedure without any cellular material. DNA was quantified with the Qubit®dsDNA HS Assay Kit and Qubit®2.0 Fluorometer (Invitrogen, Carlsbad, CA, USA). The DNA extracts were sent to Fasteris SA (Geneva, Switzerland) for amplicon sequencing
in an Illumina MiSeq sequencing platform (Illumina, San Diego, CA, USA). The V3-V4 region of the 16S rRNA gene for bacteria was amplified using the Bakt_341F (CCTACGGGNGGCWGCAG) and Bakt_805R (GACTACHVGGGTATCTAATCC) primers (Herlemann et al., 2011). Sequences, provided as pre-trimmed and pre-demultiplexed, were processed with Qiime2 (Bolyen et al., 2019) using the dada2 pipeline (Callahan et al., 2016). Sequences were truncated and joined to full denoised sequences of 464 bp. These sequences were grouped as amplicon sequences variants (ASVs). Taxonomy was assigned using
the Silva database release 132 (Quast et al., 2012) and the vSEARCH-based consensus taxonomy classifier (Rognes et al., 2016). Further analysis was performed in R Studio V3.6.3 using the R version 4.2.2 with the phyloseq package (McMurdie





and Paulson, 2016), the vegan package Oksanen et al. (2022) and the ggplot2 package Wickham (2016). In order to model metabolic capabilities of the community present, ASVs matching those present in the DNA blank extract, mitochondria, and chloroplast signals were removed from the database before exporting the representative sequences and the ASVs count table.

A taxonomy was then assigned to the PYCRUSt2 (Douglas et al., 2020) pipeline outputs using the SILVA database (Quast et al., 2012) to highlight organisms' involvement in metabolic functionalities, such as those related to acetate production or degradation. These steps were performed in R using the biomformat (McMurdie and Paulson, 2016), tidyverse (Wickham et al., 2019) and ggplot2 (Wickham, 2016) packages.

## 3 Results

### 3.1 Organic and inorganic anions

Results of the IC analyses show relatively constant $Cl^-$ and $F^-$ concentrations throughout the power plant with around $4 \, \mathrm{g \, L^{-1}}$ and $10 \, \mathrm{mg \, L^{-1}}$, respectively (Table 2). $SO_4^{2-}$ values range from $500.3$–$570.3 \, \mathrm{mg \, L^{-1}}$. Bromide concentrations were ranging from $2.5$–$15.2 \, \mathrm{mg \, L^{-1}}$. $Cl^-$ and $SO_4^{2-}$ were slightly lower with around $3.4 \, \mathrm{g \, L^-}$ and $490 \, \mathrm{mg \, L^-}$, respectively, compared to Westphal et al. (2019). Acetate was the predominant organic acid anion with concentrations ranging between $5.6$–$7.4 \, \mathrm{mg \, C \, L^{-1}}$.

Propionate was found slightly above the detection limit with $0.61 \, \mathrm{mg \, C \, L^{-1}}$ in the injection side sample from June. Formate, butyrate, valerate, and oxalate were not detected ($< 0.6 \, \mathrm{mg \, C \, L^{-1}}$). Analyses of the inhibitor showed $220.9 \, \mathrm{mg \, C \, L^{-1}}$ of acetate, leading to the assumption that with the reported addition of $10 \, \mathrm{mg \, L^{-1}}$ of inhibitor to the fluids, $2.2 \, \mathrm{\mu g \, C \, L^{-1}}$ acetate in the fluid comes from the inhibitor.

### 3.2 DOC and bulk fractions

The DOC in the fluid samples ranges from $8.4$–$10.3 \, \mathrm{mg \, C \, L^{-1}}$ (Table 2), showing a decrease along the pathway. In March, the DOC decreased from the production to the injection side whereas in June the concentrations seemed relatively uniform with slightly higher DOC in the heat central. In Westphal et al. (2019), the DOC was reported to be $14.5 \, \mathrm{mg \, C \, L^-}$ in the production sample and $4 \, \mathrm{mg \, C \, L^-}$ in the injection sample, also showing a decrease along the pathway, but a much stronger one. The inhibitor DOC comprises $102.1 \, \mathrm{g \, C \, L^-}$. With the reported dosage, the inhibitor contributes approximately $1.02 \, \mathrm{mg \, C \, L^{-1}}$ to the

total DOC of the fluids. The DOC fractions as measured by the size-exclusion-chromatography show a predominant LMWA fraction in every sample (Table 2), which can be attributed to the high acetate concentrations in the fluid. The Makro fraction is the second most abundant ($16.5$–$19.4 \, \%$), followed by the LMWN fraction ($8.5$–$11.8 \, \%$). In this study, we were not able to distinguish the Makro.1, Makro.2, and Makro.3 fractions since the chromatograms display only one peak spanning across the retention times for all three Makro fractions. This peak therefore reresents the whole Makro fraction ($10,000$–$350 \, \mathrm{Da}$). HOC

was detected only in the March samples ($12.6$–$18.8 \, \%$). The relative abundance of the LMWA fraction along the flowpath in both campaigns correlates with the respective organic acid anion trends of the samples. In the inhibitor sample, the Makro fraction accounts for $99.14 \, \%$ of the DOC. With the amount of organic carbon in the Makro fraction, as given by the SEC in





**Table 2.** DOC, concentrations of organic and inorganic anions, and relative abundance of the DOC fractions of the production (P), heat central (HC), injection (I) fluid, and inhibitor (SI) samples for the sampling campaign in March 2021 and June 2021 measured by LC-OCD and IC. Fluid data was compiled from Leins et al. (2023).

| | March 2021 | | | June 2021 | | | |
| --- | --- | --- | --- | --- | --- | --- | --- |
| | P | HC | I. | P | HC | I | SI |
| $Cl^-$ (g $L^{-1}$) | 3.92 | 4.08 | 4.03 | 4.02 | 3.85 | 3.73 | 139.1 |
| $SO_4^{2-}$ (mg $L^{-1}$) | 544.75 | 570.36 | 554.87 | 545.11 | 520.36 | 500.31 | 14.7 |
| $F^-$ (mg $L^{-1}$) | 10.08 | 10.29 | 10.15 | 10.46 | 10.55 | 10.44 | 2 |
| $Br^{-1}$ (mg $L^{-1}$) | 2.67 | 2.56 | 4.53 | 12.41 | 12.23 | 15.24 | <1 |
| | | | | | | | |
| Acetate | | | | | | | |
| (mg $L^{-1}$) | 17.15 | 16.69 | 14.19 | 17.94 | 18.19 | 17.88 | 542.9 |
| (mg C $L^{-1}$) | 6.97 | 6.79 | 5.77 | 7.29 | 7.40 | 7.27 | 220.9 |
| Propionate | | | | | | | |
| (mg $L^{-1}$) | <1 | <1 | <1 | <1 | <1 | 1.25 | <1 |
| (mg C $L^{-1}$) | | | | | | 0.61 | |
| Σ Organic acid anions | | | | | | | |
| (mg $L^{-1}$) | 17.5 | 16.69 | 14.19 | 17.94 | 18.19 | 19.13 | 542.9 |
| (mg C $L^{-1}$) | 6.86 | 6.67 | 5.67 | 7.29 | 7.4 | 7.77 | 220.9 |
| | | | | | | | |
| DOC (mg C $L^{-1}$) | 10.36 | 10.01 | 8.55 | 8.76 | 9.24 | 8.48 | 102,180 |
| **DOC Fractions** | | | | | | | |
| HOC % | 12.59 | 12.69 | 18.82 | 0 | 0 | 0 | 0 |
| Makro % | 16.48 | 16.77 | 18.9 | 16.82 | 17.15 | 19.37 | 99.14 |
| LMWA % | 59.13 | 60.98 | 53.78 | 72.15 | 71.59 | 71.55 | 0 |
| LMWN % | 11.81 | 9.56 | 8.5 | 11.03 | 11.26 | 9.08 | 0.86 |

the fluid samples, and the dosage of the inhibitor, we calculated a contribution of 60–74 % of the Makro fraction in the fluid samples to be derived from the inhibitor.

## 3.3 Molecular composition of the DOM

While LC-OCD analyses provides general information about the molecular size distribution of the DOM, FT-ICR-MS enables highly-resolved insight into the molecular composition of DOM compounds within a mass range from 150 to 1000 Da from LMWA and Makro fractions in ESI(-) and from LMWN and Makro fractions in APPI(+) mode.



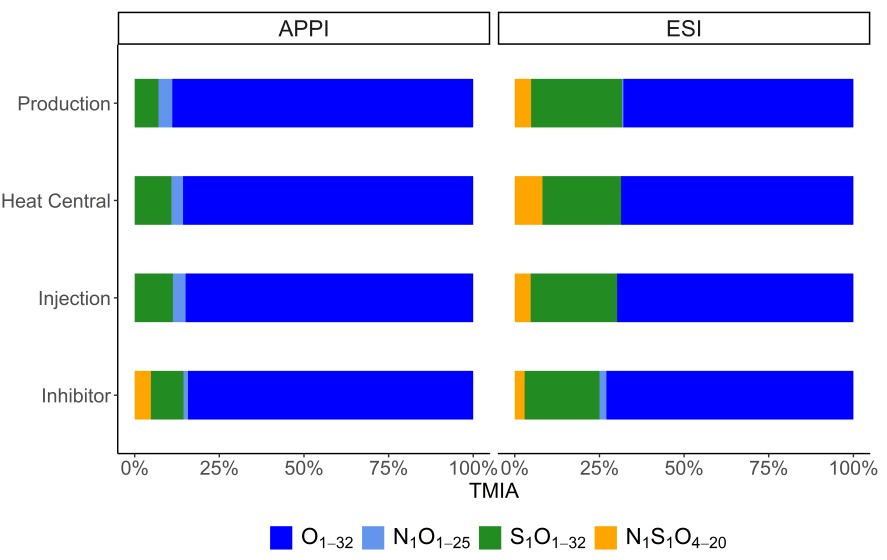

**Figure 2.** Relative abundances of assigned compound classes ($O_x$, $N_1O_x$, $S_1O_x$, and $N_1S_1O_x$) in APPI and ESI Mode.

### 3.3.1 ESI(-)-FT-ICR-MS

Negative mode ESI FT-ICR-MS of the Bad Blumau samples provided several hundred assigned formulas in the mass to charge (*m/z*) range of 157–987 (Table 3). The main compound classes in the DOM of the fluid samples were, in decreasing abundance, oxygen containing compounds ($O_x$) with around 67 % of the total monoisotopic ion abundance (TMIA), sulfur and oxygen containing compounds ($S_1O_x$) with 22.7–26.6 %TMIA, and nitrogen, sulfur and oxygen containing compounds ($N_1S_1O_x$) with 4.5–4.7 %TMIA. $N_1O_x$ compounds were only present in the production side sample (0.4 %TMIA). DOM from the inhibitor

sample shows a similar distribution with $O_x$ accounting for 72.9 %TMIA, $S_1O_x$ with 22.1 %TMIA, $N_1S_1O_x$ with 2.9 %TMIA, and $N_1O_x$ with 2 %TMIA (Fig. 2). The fluid samples are affected by inhibitor signals ranging from 31–65.3 %TMIA.

The sample with the highest numbers of assigned formulas is the heat central (Table 3). The mean number of atoms across all samples ranges from 22.7–24.2 (C), 29.3–31.9 (H), and 15.2–16.8 (O). The $M_n$ and $M_w$ range from 554.9–595 and 623.3–670.7, respectively, and DBE values are around 8.5 to 8.9 (Table 4). The $AI_{mod}$ shows that the majority of the signals are aliphatic

compounds ($AI_{mod} \leq 0.5$), with a few aromatic ($0.5 < AI_{mod} < 0.67$) and condensed aromatic ($AI_{mod} \geq 0.67$) compounds (Table 4).

The Van Krevelen diagrams were used to visualize compositional differences in the samples by presenting the molecular ratio of H/C and O/C atoms. To simplify the comparison, we followed the differentiation into four groups with different H/C and O/C ranges after Zhu et al. (2019) (I: H/C > 1 and O/C < 0.5 including lipids, proteins and part of the lignins, II: H/C > 1

and O/C > 0.5 including amino sugars, carbohydrates and part of the tannins, III: H/C < 1 and O/C < 0.5 including condensed hydrocarbons and part of the lignins and IV: H/C < 1 and O/C > 0.5 including partly condensed hydrocarbons and tannins). The majority of the signals in the fluid and inhibitor samples of the ESI data are present in area II (Fig.3)





**Table 3.** Total and unique numbers of assigned monoisotopic signals within their sample group including the distribution of the main elemental classes, total mass range, percentage of TMIA derived from inhibitor signals, elemental numbers (carbon, hydrogen, oxygen), and molecular weight.

| Well | No. of formulas | | | | | | Mass range | SI amount | Mean | | | $M_n$ | $M_w$ |
|---|---|---|---|---|---|---|---|---|---|---|---|---|---|
| | Total | $O_x$ | $N_1O_x$ | $S_1O_x$ | $N_1S_1O_x$ | Unique | (Da) | (TMIA %) | C | H | O | Total | Total |
| **ESI negative** | | | | | | | | | | | | | |
| P | 669 | 521 | 5 | 80 | 63 | 106 | 171–995 | 65.3 (280) | 23 | 29.3 | 16.8 | 582.1 | 629.6 |
| HC | 809 | 619 | 0 | 82 | 108 | 198 | 179–981 | 31.1 (241) | 24.2 | 31.9 | 16.4 | 595 | 639.9 |
| I | 573 | 469 | 0 | 61 | 43 | 16 | 165–987 | 42.4 (222) | 22.9 | 29.8 | 15.9 | 569.3 | 623.3 |
| SI | 472 | 409 | 7 | 40 | 16 | 148 | 157–987 | - | 22.7 | 29.9 | 15.2 | 554.9 | 670.7 |
| **APPI positive** | | | | | | | | | | | | | |
| P | 700 | 594 | 55 | 51 | 0 | 46 | 209–1019 | 64 (496) | 20.8 | 24.5 | 10 | 436.6 | 467.6 |
| HC | 649 | 547 | 39 | 63 | 0 | 52 | 193–1019 | 58.1 (460) | 21.2 | 25.8 | 9.74 | 440.6 | 473.6 |
| I | 649 | 555 | 43 | 50 | 0 | 40 | 209–912 | 57.5 (463) | 21 | 25 | 9.52 | 433.8 | 461.3 |
| SI | 1741 | 1276 | 82 | 189 | 194 | 1262 | 181–1175 | - | 22.7 | 28.8 | 12.8 | 511.9 | 543.2 |

$M_n$: number-averaged molecular weight; $M_w$: weight-averaged molecular weight

### 3.3.2 APPI(+)-FT-ICR-MS

APPI FT-ICR-MS of the Bad Blumau fluid samples provided assigned formulas in the *m/z* range of 193–1019 (Table 3). The

main compound classes in the DOM of the fluid samples were, in order of predominance, $O_x$ with 69–76.1 %TMIA, $S_1O_x$ with 6–9.2 %TMIA, and $N_1O_x$ 2.8–3.5 %TMIA. For the inhibitor sample, 1741 signals were assigned with a similar compound class distribution of $O_x$ with 81.8 %TMIA and $S_1O_x$ with 9.3 %TMIA. $N_1S_1O_x$ compounds were assigned with 4.7 %TMIA. $N_1O_x$ compounds have the lowest abundance with 1.3 %TMIA (Fig.2). The sample with the highest numbers of assigned formulas is the inhibitor (Table 3). Comparing only the fluid samples, the heat central has slightly more unique formulas. The

mean number of atoms across all samples ranges from 20.8–22.7 (C), 24.5–28.8 (H), and 9.5–12.8 (O). The $M_n$ and $M_w$ range from 433.8–511.9 and 461.3–543.2, respectively. The DBE values are found around 8.8 to 9 and $AI_{mod}$ around 0.23 (Table 4). The Van Krevelen diagrams show a dominant distribution of the signals within area I and II (Fig.3). $N_1S_1O_x$ compounds are only present in the inhibitor and were not detected in the fluid samples.





**Figure 3.** Van Krevelen diagrams of the fluid and inhibitor samples in (a) ESI(-) and (b) APPI(+) mode color-coded by the assigned compound classes.





### 3.4 Microorganisms in the fluids

The composition of the bacterial communities in the fluids in the three samples in which DOC was characterized was investigated by amplicon sequencing of the 16S rRNA. The relative abundance of the 10 most abundant bacterial phyla present in the different fluids from Bad Blumau is shown in Fig. 4. A very significant shift in community composition was detected consistently in the replicate samples of the three sampling points. In all three, the Firmicutes phylum was dominant. Amplicon sequence variants (ASVs) assigned to Proteobacteria were present in a small relative abundance in the production well,

but became highly abundant (co-dominant with ASVs from Fimicutes) at the heat central. However, at the injection site, Proteobacteria was not detected and was replaced by the *Thermotogae* phylum, which dominated the community together with Firmicutes. The *Thermotogae* phylum was also detected in all the replicates from the heat central, but at a much lower relative abundance. At the production site, besides the dominant phyla, there was also a low abundance of *Actinobacteria*, *Cyanobacteria* and *Planctomycetes*, followed by few *Nitrospirae* (in one sample), *Thermotogae* (in one sample) and *Verrucomicrobia*. At

the heat central, there was a low abundance of *Thermotogae*, *Actinobacteria*, and *Nitrospirae* while at the injection side, there was a low abundance of *Thermotogae*, as well as an extremely low relative abundance of *Actinobacteria* and *Bacteroidetes*. The same analysis was performed at a lower taxonomic rank. At the genus level, the drastic changes in the composition of the bacterial community in the different sampling points were more easily observed. The relative abundance of the 10 most abundant genera per sampling point is represented in Fig. 5. At the production well, the genus *Bacillus* (Firmicutes) dominated

the fluids. At the heat central, the *Caulobacter* (Proteobacteria) and SCADC1-2-3 (Firmicutes) genera were dominant, while at the injection site the genus of sulfate-reducing bacteria (SRB) *Desulfotomaculum* (Firmicutes) was dominant. However, it is important to note that the top five genera at each sampling point does not represent the entire community, but only a subfraction of the most abundant members of the community. At the production well and the heat central, the top five genera represent around 60 % of the detected community, while at the injection site, the top five genera represent around 75 % of the community.

This suggest that the community at the injection site was less diverse than the communities present at the two other sites.

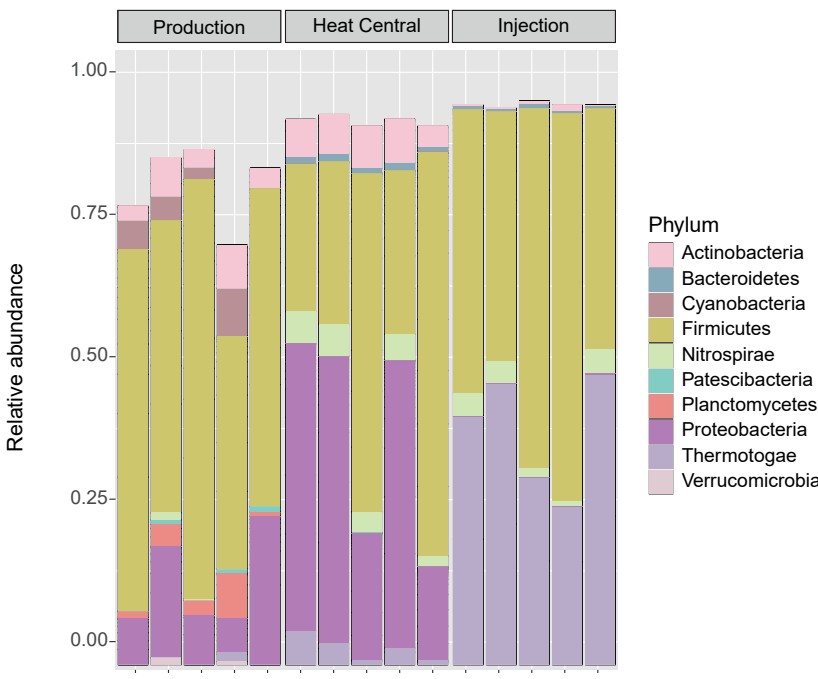

**Figure 4.** Top 10 bacterial phyla in relative abundance (in all samples). The ASVs not assigned to any phylum (unclassified) are not shown.



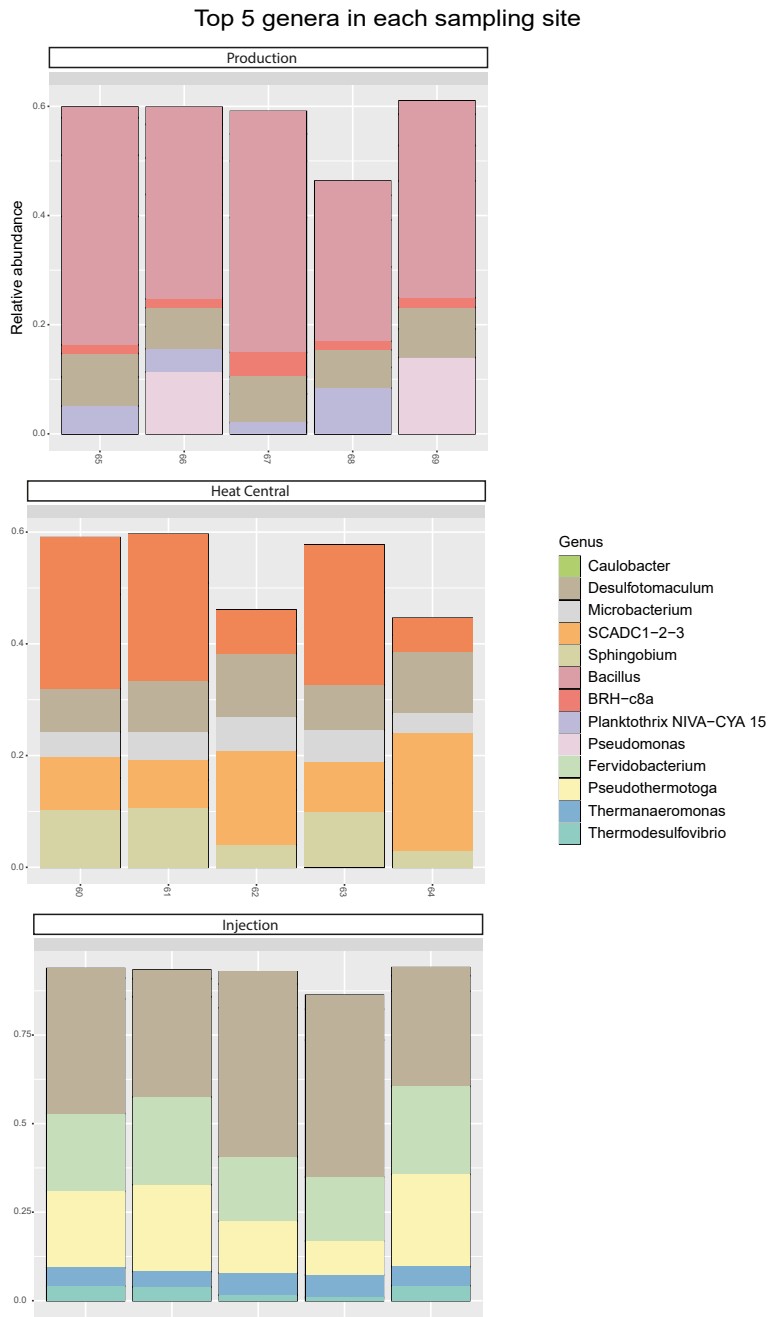

**Figure 5.** Top 5 bacterial genera in relative abundance (calculated for each sampling point separately). The ASVs not assigned to any phylum (unclassified) are not shown.





Based on the phylogenetic composition, the metabolic potential of the bacterial communities in Bad Blumau was predicted. Metabolic pathways were predicted using PICRUSt2 (Douglas et al., 2020), which predicts the metabolic capacities of a given bacterial genus based on the conserved V3-V4 regions of the bacterial 16S rRNA gene. After predicting the pathways that can be present, the software also predicts the contribution of bacteria from a given genus to the predicted pathway. The potential metabolisms resulting in acetate production were the fermentation of lysine to acetate and butyrate, the fermentation of pyruvate to acetate and lactate, and the fermentation of hexitol, which can also lead to the production of lactate, formate, ethanol, and acetate (Figure 6). Acetate can then be used in methanogenesis (Figure 6). At the production site, Bacillus is the dominant genus contributing to the different metabolic pathways identified here, while at the injection site, *Desulfotomaculum* was the dominant genus. The heat central, depending on the targeted metabolism, either represented a transition between the production and injection points, (Figure 6) or can be dominated by a different genus, like the *Caulobacter* genus in the case of the lysine fermentation (Figure 6).



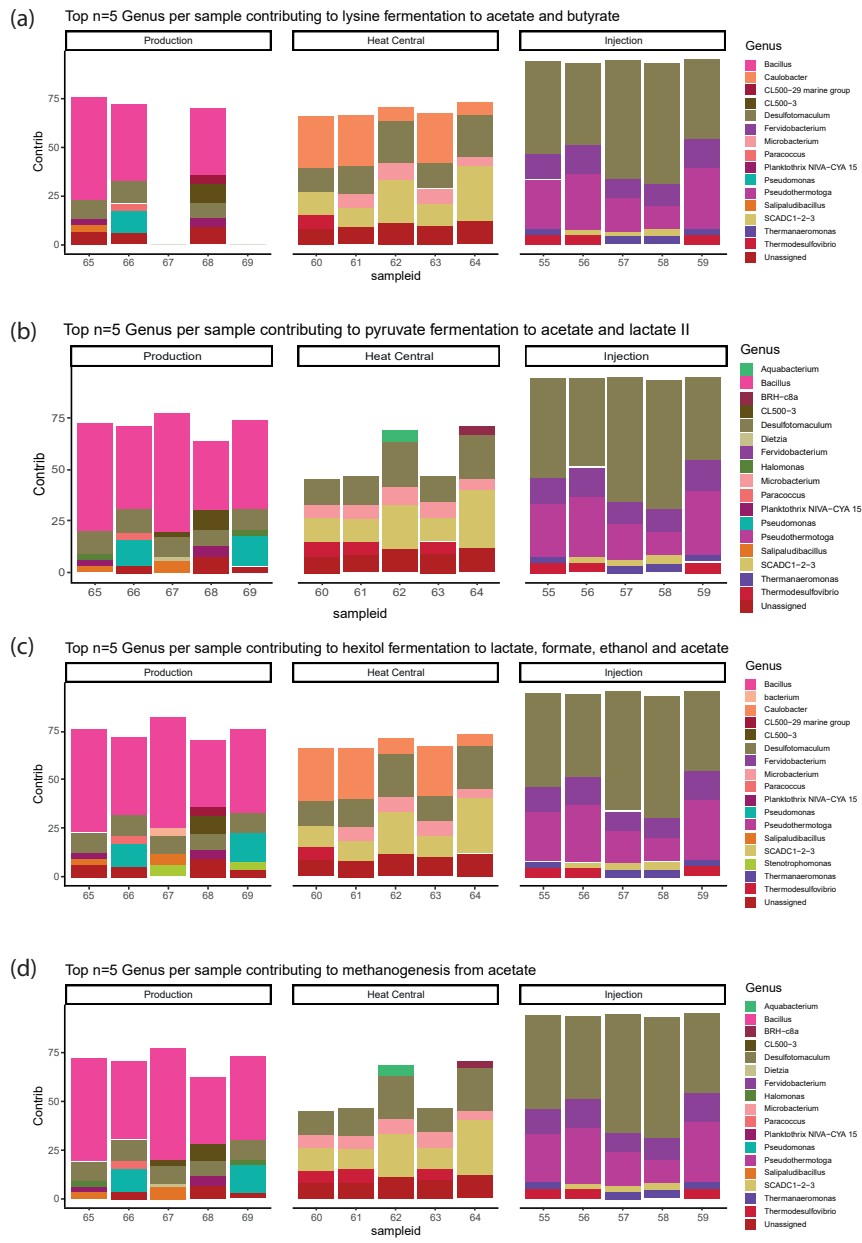

**Figure 6.** Top 5 Genus per sample contributing to (a) lysine to acetate and butyrate fermentation, (b) pyruvate fermentation to acetate and lactate II, (c) hexitol fermentation to lactate, formate, ethanol and acetate, (d) methanogenesis from acetate.





## 4 Discussion

LC-OCD analyses provided a good general overview of the DOC fractions showing that most of the DOM is present in the LMWA fraction. IC results confirmed that this acid fraction is mainly composed of acetate and with FT-ICR-MS we were

able to identify the inhibitor-derived organic compounds in the fluids. This arises the questions: (1) is the distinction and quantification between synthetic and natural organic matter possible? (2) Why does DOM and microbial diversity change along the fluid pathway and is there a correlation between the two? And (3) Where does the acetate derive from and what is the metabolic pathway for it? These three questions are discussed in the following.

### 4.1 DOM composition and impact of the inhibitor in the fluid of the deep geothermal site Bad Blumau

The TMIA of the formulas that derive from the inhibitor was added up for each fluid sample to determine the abundance of inhibitor-derived compounds in the fluids. This abundance of inhibitor derived compounds decreases from the production side (65.3 %TMIA) to the heat central (31.1 %TMIA), followed by an increase to the injection side (42.4 %TMIA) in ESI(-) mode (Table 3). The overall decrease in abundance of inhibitor derived signals could be an indication for degradation, alteration or dilution of the inhibitor along the fluid pathway. In the fluid samples (P, HC, I), the $O_x$ class (x = 1–32) shows a roughly

Gaussian distribution of the relative abundances, dominating in the range of $O_{12-24}$, especially the even numbered compounds (Fig. 7). In the $S_1O_x$ class, the odd numbered oxygen compounds ($S_1O_{13-27}$) exhibit a significantly higher relative abundance compared to the other signals. These distinct peaks coincide with the dominant peaks in the inhibitor sample. $S_1O_x$ compounds in the inhibitor sample are mainly composed of compounds with odd numbered oxygen numbers in the range of $S_1O_{13-27}$. This suggests that $S_1O_{13-27}$ compounds derive mainly from the inhibitor. The assigned signals from the fluid and inhibitor samples

show a strong overlap between inhibitor and production side sample with up to 65.3 %, further reinforcing this assumption (Fig. 7). Both the heat central and injection side samples do not show such a strong overlap. Nevertheless, it is likely that these $S_1O_x$ compounds derive from the inhibitor and had undergone chemical alteration along the flow path, as to which the signals from the fluids and inhibitor would no longer match. A similar result can be seen in the $O_x$ compound class after subtracting the inhibitor signals. The few $N_1O_x$ compounds all have odd oxygen numbers as the inhibitor. These compounds were all

introduced by the inhibitor and are only present in the production side sample. $N_1S_1O_x$ compounds, seem to be less affected by the inhibitor. Mainly $N_1S_1O_4$ and likely the even numbered oxygen compounds in the range of $N_1S_1O_{8-20}$ are introduced by the inhibitor. The intensity-weighted averages of the molecular H/C and O/C ratios of all detected formulas compared to the inhibitor subtracted signals, show a stronger difference of O/C values in the production side. This might be explained by the $S_1O_{13-27}$, which were mainly identical to inhibitor signals in the production side but not in the heat central and the injection

side (Table 4). Approximately half of the signals in the fluid samples in area II of the Van Krevelen diagrams match with the signals of the inhibitor (Fig. 8). With the exception of the production side sample, the inhibitor signals in the fluid samples in area I are mainly present at lower H/C (< 1.6) values. Two groups of $S_1O_x$ signals are mainly present in the production side. One with high H/C values around 1.6 and one with H/C around 0.8 and both groups at approximately O/C 0.2. These signals


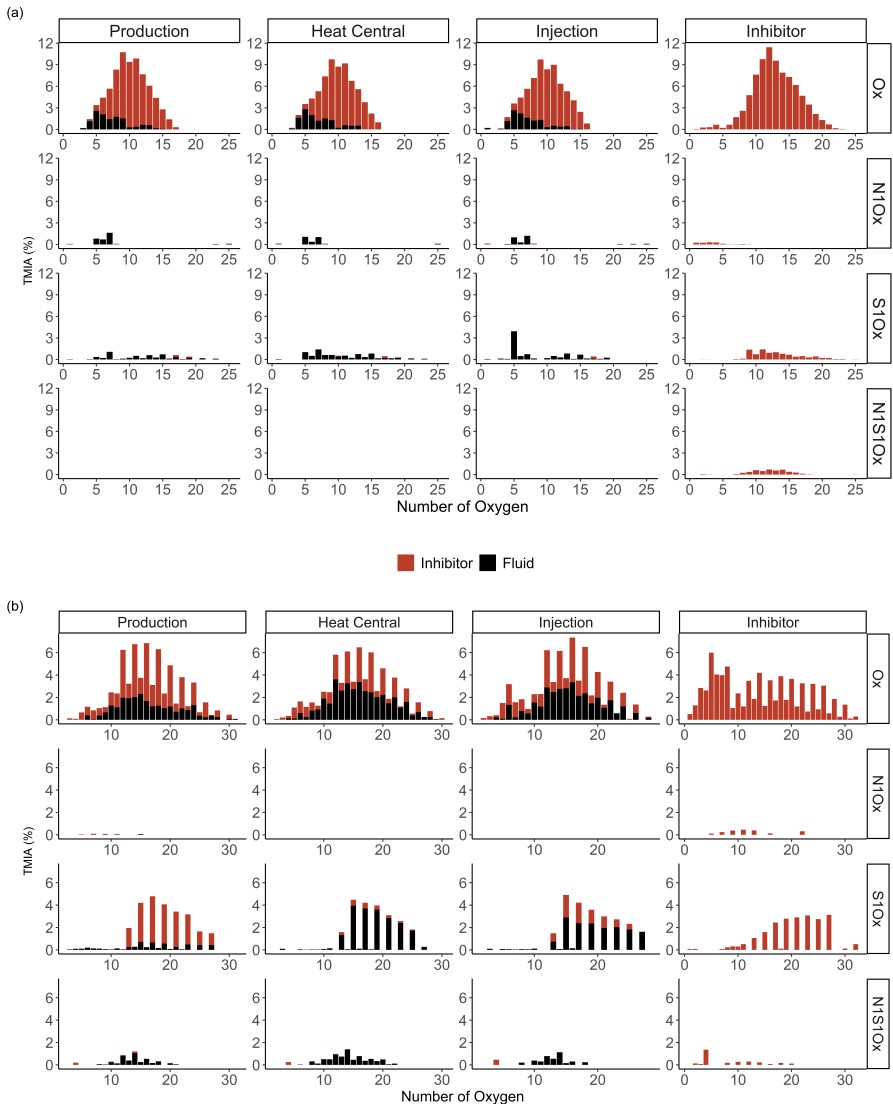

**Figure 7.** TMIA vs. number of oxygen for every compound class and sample in (a) APPI(+) and (b) ESI(-) mode. Data colored in red within the fluid samples represent the fluid signals that match with the signals found in the inhibitor sample.

correspond to $S_1O_x$ ($x < 11$). This further suggests that these compounds may not derive from the inhibitor but are part of the

natural DOM or are coming from other sources.

Similar to the ESI(-) dataset, the APPI(+) data shows that the proportion of the TMIA in the fluid samples deriving from the inhibitor decreases from the production to the injection side. Here, the inhibitor derived TMIA shows a minimal and gradual decrease over the three sampling points from 64 % to 57.5 % (Table 3). The mean oxygen numbers decrease from the production to the injection side and differ clearly from the mean oxygen number of the inhibitor. The mean number of





carbon and hydrogen, as well as the mean $M_n$, and $M_w$, increase from the production side to the heat central, followed by a slight decrease to the injection side. An opposite trend is shown for the mean DBE values (Table 4). In terms of mean DBE, carbon, and hydrogen number, the production side sample is most similar to the inhibitor sample. The $O_x$ class shows a Gaussian distribution centered around $O_{9-12}$ in the fluid samples, and $O_{11-13}$ in the inhibitor sample (Fig. 7). A comparison of the assigned $O_x$ signals in the fluids with those in the inhibitor show that a large amount with up to 63.2 % of the signals in the

fluids are introduced by the inhibitor. Removing the inhibitor signals results in a shift of the $O_x$ distribution to $O_{3-9}$ centered around $O_5$ in the fluid samples. $O_x$ compounds with more than 15 oxygen atoms are solely found in the inhibitor (Fig. 7). The introduction of molecules with high numbers of oxygen is also shown by the average O/C ratios decreasing if the inhibitor signals are removed from the data (Table 4).

The APPI data shows distinct patterns for the fluid and the inhibitor samples in the Van Krevelen diagrams (Fig. 8). The

majority of the signals in the fluid samples are accumulated in area I, while the inhibitor sample shows a strong accumulation in area II and a weaker accumulation in area I at O/C < 0.2. Signals that were only found in the fluid samples are mainly present in the O/C and H/C range of 0.2–0.4 and 1–1.7, respectively. They represent $O_x$ as well as distinct $N_1O_x$ and $S_1O_x$ compound classes compared to the inhibitor. Typically, signals in this range are attributed to compounds deriving from proteins (Sleighter and Hatcher, 2007). Since microorganisms were reported for the Bad Blumau fluids (Westphal et al., 2019), these

signals may represent microbial activity. Generally, signals within the lipid group of the van Krevelen diagrams (H/C around 2 and O/C below 0.2) represent compounds deriving from cell membranes of microorganisms (Sleighter and Hatcher, 2007) and are therefore also good indicators for microbial activity. Especially, the heat central sample shows an accumulation of signals in this area. It was assumed that the microorganisms in the Bad Blumau fluids were likely to feed on macromolecular organic matter introduced by the scaling inhibitor (Westphal et al., 2019). FT-ICR-MS data shows certain compound classes that are

present in the inhibitor sample but absent in the fluid samples (Fig. 4). This is shown in the APPI(+) $N_1O_x$, $S_1O_x$, and $N_1S_1O_x$ as well as ESI(-) $N_1O_x$, and $N_1S_1O_x$ compound classes (Fig. 5). The absence of these inhibitor specific compounds in the fluid samples could be explained by: (1) alteration in form of chemical reactions with the fluid forming complexes, (2) degradation after the injection into the fluids, (3) no detection in the fluid samples due to strong dilution of the inhibitor, (4) degradation by microorganisms that target these specific compounds.

The major absence of signals in area III and IV of the van Krevelen diagrams in both ionization modes suggests a low amount of highly condensed and aromatic compounds (Kim et al., 2003; Sleighter and Hatcher, 2007). For example the DOM of a dismal swamp presented by ESI FT-ICR-MS (Sleighter and Hatcher, 2007) and pore water of a Mangrove (Koch et al., 2005) reported a wider range of signals with H/C ratios below 1 compared to the samples in this study. A strongly different distribution of DOM signals was reported for coal water extracts (kerogen type III) where the majority of the signals was found

at H/C ratios below 1 in area III and IV (Zhu et al., 2019).

Both ionization modes show only a few aromatic compounds that derive from the inhibitor. Those deriving from the inhibitor however, have a notable effect on the TMIA in the APPI(+) dataset since aromatic abundance is decreasing in the inhibitor filtered data (Table 4). Aromatic and condensed aromatic compounds may be of natural origin from the reservoir or from other sources. For example, aromatic $S_1O_x$ compounds detected by ESI(-) mode in the fluid samples could be polystyrene-,





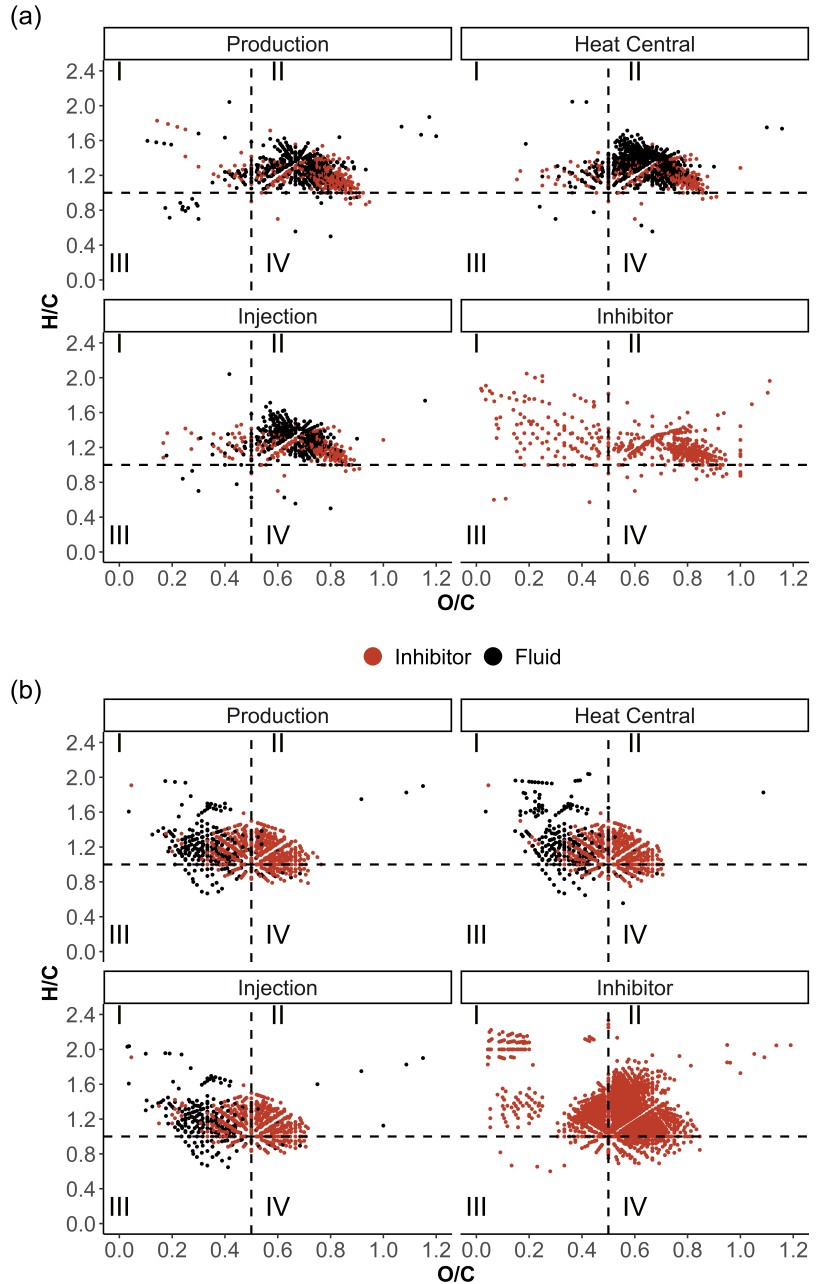

**Figure 8.** Van Krevelen diagrams of the fluid and inhibitor samples in (a) ESI(-) and (b) APPI(+) mode color-coded by matching signals in inhibitor and fluid samples.

naphthalene-, and lignosulfates which are used as superplasticizers in cement (Flatt and Schober, 2012; Hewlett et al., 2019) and





**Table 4.** Comparison of the average and standard deviations of intensity-weighted averages of DBE, AI$_{mod}$, H/C, and O/C ratios of all detected APPI(+) and ESI(-) signals and signals where the inhibitor was removed from the dataset (w/o Inh.). The proportion of aromatic compounds is given as %TMIA.

| | ESI | | | | | | APPI | | | | | |
| --- | --- | --- | --- | --- | --- | --- | --- | --- | --- | --- | --- | --- |
| | Production | | Heat Central | | Injection | | Production | | Heat Central | | Injection | |
| | All | w/o Inh. | All | w/o Inh. | All | w/o Inh. | All | w/o Inh. | All | w/o Inh. | All | w/o Inh. |
| H/C ratio | 1.26 | 1.29 | 1.31 | 1.36 | 1.29 | 1.33 | 1.17 | 1.23 | 1.20 | 1.30 | 1.18 | 1.22 |
| O/C ratio | 0.73 | 0.68 | 0.67 | 0.65 | 0.69 | 0.67 | 0.49 | 0.35 | 0.47 | 0.33 | 0.46 | 0.33 |
| DBE | 8.93 | 8.9 | 8.75 | 8.62 | 8.47 | 8.58 | 9.05 | 9.7 | 8.85 | 9.07 | 9.05 | 9.78 |
| AI$_{mod}$ | 0.04 | 0.04 | 0.04 | 0.01 | 0.05 | 0.02 | 0.22 | 0.24 | 0.23 | 0.21 | 0.24 | 0.23 |
| Aromatics % | 0.9 | 0.8 | 0.7 | 0.5 | 1.6 | 1.3 | 3 | 1.6 | 3 | 1.6 | 2.6 | 1.8 |

could derive from the casing of the borehole. This data suggest an absence or only very low impact of petroleum hydrocarbons on the fluids since only a low proportion of DOM signals are present in area III and IV. Deep fracture water DOM from the Witwatersrand Basin reported a complete lack of signals in area III and IV, which was explained by low contribution of hydrocarbons from the organic rich reefs of the Witwatersrand Supergroup (Kieft et al., 2018).

A major difference of the Bad Blumau fluid samples compared to other natural DOM detected by ESI(-) mode is shown by the intensity-weighted average O/C values. The Bad Blumau samples exhibit high average O/C and H/C values from 0.65 to 0.68 and 1.26 to 1.33, respectively. Average values for marine, pore and river waters were reported in the O/C and H/C range of 0.32 to 0.52 and 1.13 to 1.29, respectively (Koch et al., 2008; Sleighter and Hatcher, 2008). Studies investigating the DOM of hydrothermal fluids reported average O/C values of 0.25 to 0.35 and generally high H/C ratios above 1.35 (Noowong

et al., 2021; Gomez-Saez et al., 2016). Higher O/C values are generally attributed to tannine like compounds in area II and IV at O/C greater 0.5 and H/C around 1 (Sleighter and Hatcher, 2007) originating from terrestrial plant matter and some algae (De Leeuw and Largeau, 1993). Compound types deriving from algal detritus and/or microbial biomass in marine sediments are however reflected by lower O/C and higher H/C ratios (Sleighter and Hatcher, 2008), which is shown in our APPI(+) data but not the ESI(-) results. Since the reservoir rock of Bad Blumau was formed during Palaeozoic reef development a contribution

from higher plants is unlikely. However, deuterium and oxygen isotope data indicate an influence of meteoric water for the deep thermal Bad Blumau fluids (Goldbrunner, 2000). The higher O/C ratios in the ESI(-) data could be the result of terrestrial DOM being transported into the subsurface. However, within a geothermal setting the DOM likely undergoes thermal degradation. Elevated temperature experiments with marine DOM showed a preferential loss of high-molecular weight and oxygen rich molecules within two weeks of run-time (Hawkes et al., 2016). The experiments were conducted in the temperature range of

100–380 °C and suggested that abiotic hydrothermal alteration may start at temperatures above 68 °C. It is unlikely, that at the Bad Blumau reservoir temperatures of 124 °C the intensity-weighted average O/C ratios calculated for the ESI(-) formulas would show such high values for DOM originating from the reservoir fluids. Another explanation for the still high O/C ratios





could be the strong influence of artificial DOM, even after filtering all signals deriving from the inhibitor. Contrary to the ESI(-) data, the inhibitor filtered APPI(+) data shows average O/C and H/C values from 0.33 to 0.35 and 1.23 to 1.3, respectively,

which are in good agreement with hydrothermal vent DOM detected by ESI(-) mode (Noowong et al., 2021; Gomez-Saez et al., 2016). The average DBE values of both APPI(+) and ESI(-) measurements are generally in the same range as seawater and diffuse hydrothermal fluid DOM with temperatures up to 170 °C (Noowong et al., 2021; Gomez-Saez et al., 2016). Hotter fluids (>300°C were reported with much higher average DBE values. The low grade of aromaticity in the Bad Blumau DOM with average $AI_{mod}$ values below 0.25 coincide with seawater, hydrothermal vent, and thermally altered marine DOM (Noowong

et al., 2021; Gomez-Saez et al., 2016; Hawkes et al., 2016). It was stated that the elevated temperatures at hydrothermal conditions likely resulted in a decrease of molecular formulas detected by FT-ICR-MS due to a loss of thermally unstable DOM (Noowong et al., 2021; Hawkes et al., 2016; Rossel et al., 2017; Longnecker et al., 2018). This assumption could explain the relatively low number of formulas in the Bad Blumau fluid DOM ($< 10^3$), since a similar range of detected formulas was reported for hydrothermal DOM (Noowong et al., 2021).

**4.2 Variation of the composition of DOM and microbial community along the flow path**

The DOM composition does not show much variation along the flowpath. Most notable are the signals in area I and III of the production side sample at H/C 1.7 and 0.8 in ESI(-) mode and the cluster in area I of the heat central at H/C 1.7 in APPI(+) mode (Fig. 8). The ESI(-) signals, as previously discussed, are either $N_1O_x$ compounds that derive from the inhibitor or $S_1O_x$ that are likely introduced from other artificial sources. The distinct APPI(+) signals in the heat central are solely from the

inhibitor. It has to be taken into consideration that the samples from the three sampling points do not represent the same body of water. Thus, slight variations and differences such as inhibitor signals being present in the heat central and not in the other samples do not necessarily represent processes linked to the flowpath.

In terms of the DOC concentration however, a decrease along the flowpath was observed not only in this study but also in Westphal et al. (2019). This DOC decrease could be indicative for microbial degradation. Especially, since a variety of

microorganisms were detected in this study. At the phylum level, the detection of Firmicutes, Proteobacteria and *Thermotogae* under the extreme environmental conditions in the three sampling points is not unlikely. Indeed, Firmicutes are known to form endospores (Cano and Borucki, 1995; Nicholson et al., 2002), which are highly resistant structures known to withstand conditions such as those in the power plant. Members of the Firmicutes (Filippidou et al., 2016) and Proteobacteria (Dib et al., 2008) have been detected in several extreme environments, showing their potential to withstand the environmental

conditions in deep geothermal reservoirs. Members of the phylum *Thermotogae* can be either mesophilic, thermophilic and hyperthermophilic, and most of the cultivated representatives have been obtained from extreme environments (Bhandari and Gupta, 2014).

At the production site, in addition to *Bacillus*, some *Pseudomonas*, *Desulfotomaculum*, *Planktothrix*, and ASVs related to the clade BRH-c8a were also detected. Species of *Bacillus* spp. are able to form highly resistant endospores (Nicholson et al.,

2000). Thus, the dominance of *Bacillus* at the production site, where the temperature is higher than at the other sampled sites, is not surprising. A previous bacterial diversity performed in Bad Blumau (Westphal et al., 2019) already reported the detection



of members of the genus *Desulfotomaculum*. Moreover, this genus has been detected in other geothermal systems, such as different terrestrial hot springs (Amin et al., 2013; Poratti et al., 2016) or in geothermal ground water (Daumas et al., 1988). Thus, finding a *Peptococcaea* closely related to *Desulfotomaculum* (BRH-c8a) (Sousa et al., 2018) in a geothermal system is

also not surprising. Representatives of the genus *Pseudomonas* have been reported in the oxic zone of a geothermal systems (Burté et al., 2019). The presence of Cyanobacteria belonging to the genus *Planktothrix* is more surprising, as Cyanobacteria, which are known for their phototrophic metabolism would need at least an occasional light exposure for active growth (Puente-Sánchez et al., 2018) However, Cyanobacteria have recently been found in different environments that are not exposed to light, such as the deep subsurface or around hydrothermal vents (Hubalek et al., 2016; Puente-Sánchez et al., 2018; Chen et al.,

2022). Moreover, Cyanobacteria are commonly found in geothermal environments (Ward et al., 2012). This indicates that the presence of Cyanobacteria in different ecosystems, such as deep geothermal system, is possible.

At the heat central, the dominant genera corresponded to *Caulobacter*, *Desulfotomaculum* (also detected in the production and injection sites), *Sphingobium*, SCADC1-2-3, and *Microbacterium*. Members of the genus *Caulobacter* and *Sphingobium* were detected in the oxic zone and the anoxic zone, respectively, of the same geothermal system in which *Pseudomonas* was

reported (Burté et al., 2019). The SCADC1-2-3 is a group of uncultured organisms that belongs to the *Desulfisporaceae* family (Gavrilov et al., 2022). The SCADC1-2-3 group is part of a family of thermophilic sulphate-reducing bacteria and has been detected in subsurface waters (Gavrilov et al., 2022). The *Microbacterium* species described so far are known to be tolerant to extreme conditions, for instance to the presence of arsenic (Achour-Rokbani et al., 2010), and have been isolated notably from a heated aquifer bore well (Adelskov and Patel, 2017), surface hot springs (Mehetre et al., 2019), and from the Atacama Desert

(Mandakovic et al., 2020).

At the injection well, in addition to *Desulfotomaculum*, which was highly dominant, *Fervidobacterium*, *Pseudothermotoga*, *Thermanaeromonas* and *Thermodesulfovibrio* were detected, the last two at a low relative abundance. The described species belonging to the genus *Fervidobacterium* are all anaerobic and extremely thermophilic, fermenting glucose to acetate and reducing sulfur to H$_2$S (Huber and Stetter, 2015). *Pseudothermotoga* are known thermophilic and anaerobic bacteria, isolated

from hot springs and oil reservoirs (Farrell et al., 2021). *Thermanaeromonas* species are thermophilic anaerobes able to form spores (Mori and Hanada, 2015). They were notably isolated from a geothermal aquifer (Mori et al., 2002) and from a subterranean clay environment (Gam et al., 2016). *Thermodesulfovibrio* are also anaerobic thermophilic organisms, which are able to oxidize organic substrates to acetate (Maki, 2015) and have been isolated from hydrothermal vent waters (Henry et al., 1994) or hot springs (Sonne-Hansen and Ahring, 1999).

The results of our study are largely consistent with those of a previous study investigating the bacterial diversity in the fluids collected in 2011 and 2013 at Bad Blumau (Westphal et al., 2019). In this previous study, samples were also taken at the production well and at the injection well. The third sampling point was located after CO$_2$ extraction, but before the heat central, while in our case, our third sampling point was located at the heat central itself. Nevertheless, some differences were detected. In Westphal et al. (2019), the classes alpha-proteobacteria and beta-proteobacteria were found in the production fluids. In our

case, alpha-proteobacteria were detected in the production fluids and after the heat central, but no beta-proteobacteria were detected as part of the most abundant classes (data not shown). *Clostridia* were detected in all fluids by Westphal et al. (2019),





which was our case as well. At the injection well, the classes Actinobacteria, Clostridia and Thermotogae were the dominant classes detected by (Westphal et al., 2019). The Clostridia class was detected in the different samples in both studies, and the Clostridia and Thermotogae classes become dominant at the injection well displacing Actinobacteria, which were only

present in a low relative abundance. Our observations also confirm the presence of the Bacilli class as a dominant class in the production site. Previously, the Ignavibacteria and the Nitrospira class were detected in the injection fluids, but in our case, Ignavibacteria were detected in a very low abundance in the fluids from the heat central and the injection well and the Nitrospira were not detected. However, the class Thermodesulfovibriona was detected in low abundance at the heat central and the injection fluids and this class was previously part of the Nitrospira class (Rabus et al., 2015; Umezawa et al., 2021). Thus,

the difference concerning the Nitrospira class in our analysis compared to the analysis made by Westphal et al. (2019) may only reflect a change in the bacterial taxonomy.The genus *Desulfotomaculum* was consistently detected at the production well, after $CO_2$ removal and at the injection well in both studies. At the production well, the study by Westphal et al. (2019) and the present study detected the class Bacilli and the genus *Pseudomonas*. Several genera, such as *Comamonas* and *Ralstonia*, were detected in the production fluids by Westphal et al. (2019), but were not among the top 5 genera observed in our study. Westphal

et al. (2019) also detected *Thermodesulfovibrio*, *Desulfovirgula*, *Desulfovibrio*, *Fervidobacterium* and *Thermanaeromonas* at the injection well. In our case, *Thermodesulfovibrio*, the *Fervidobacterium* and *Thermanaeromonas* genera were also detected. One difference between the two studies is the detection of Cyanobacteria (*Planktothrix*) and *Planctomycetes* at the production well, but none of these phyla are dominant.

Overall, both studies highlight the changes of the bacterial communities along the power plant system, from the production

well to the injection well. Such differences in the communities highlight the impact of the conditions in the fluids on the bacterial communities, with some bacteria replacing others as the conditions change within the system (Alawi et al., 2011; Lerm et al., 2013; Westphal et al., 2019). Moreover, this is consistent with the tests to change the bacterial communities by adding nitrate in the fluids, which led to changes in the community. More importantly, the consistency between two independent studies, demonstrates that the communities are relatively stable over several years. The differences in the results may either

originate from differences in the detection methods, or from small changes in the community. The dominant genera were nevertheless the same as the ones that were detected previously, and the changes in the communities from the production well to the injection well followed the same patterns observed by Westphal et al. (2019). This confirms that the presence of these microorganisms in the Bad Blumau system was not only temporary, as a by-product of maintenance work for instance. Moreover, despite a relatively high flow rate, changes in the microbial community show that these conditions are compatible

with a reactivation and development of a microbial community.

## 4.3   The role and origin of acetate: link to active microbial community

Acetate forms the dominant part of the DOC in the fluid samples. Several possible sources were considered for its presence in our samples: abiotic origin (e.g. water-oil contact or inhibitor derived) and biotic as a byproduct from active microbial communities. Water-oil contact is a known source for organic acid anions in the fluid since it was suggested to increase the

LMWA content due to the release of hydrophilic acids (Reinsel et al., 1994). Oil-field waters are generally described to contain





predominantly acetate, but also detectable amounts of formate, propionate, butyrate, and valerate (Carothers and Kharaka, 1978; Hatton and Hanor, 1984; Kharaka et al., 1985, 1997). However, the lack of butyrate and valerate as well as only sporadic detection of propionate in the Bad Blumau samples make it less likely that the organic acid anion content is associated with the water-oil contact.

In deep fracture waters (Sherwood Lollar et al., 2021) thermodynamic conditions were found to be favorable for abiotic generation of formate and acetate, however, at temperatures (25 °C) significantly different compared to Bad Blumau. Abiotic production of acetate has also been shown to occur at a temperature of 60 °C at 2 bar and the presence of a greigite (Fe$_3$S$_4$) catalyst (Preiner et al., 2020). However, abiotic production of acetate due to water-rock reactions seems to be correlated with the presence of formate (Kieft et al., 2018; Sherwood Lollar et al., 2021; McDermott et al., 2015; Lang et al., 2010), which

is absent in the Bad Blumau samples. Since no formate was detected in the Bad Blumau samples it is unlikely that abiotic reactions have a significant effect on the production of the acetate.

Acetate is therefore most likely of biogenic origin and formed as primary breakdown product of complex organic matter due to microbial degradation by fermentative anaerobes. Metabolic pathways were predicted in our study using PICRUSt2 (Douglas et al., 2020) based on the community composition. These predictions do not necessarily prove the existence of a given

metabolism, but can be a useful aid to guide future enrichment or the selective detection of specific microbial groups/metabolic processes. As acetate was present in Bad Blumau in high amounts (14.2–18.2 mg L$^{-1}$), all the pathways linked to acetate were assessed (Fig. 6). Many microorganisms have either the potential to use acetate, or to produce it. We show here that this is also the case within the Bad Blumau fluid systems, where the production and use of acetate by microorganisms would be possible at the production well, the heat central, and the injection well.

Relatives of the fermentative bacteria (*Thermoanaerobacter brockii*) were detected in the produced fluids by Westphal et al. (2019), which are described to produce lactic acid, acetic acid, H$_2$, and CO$_2$ as fermentation products. Macromolecular components from the scaling inhibitor would likely act as energy and carbon source for this fermentation process, as was already suggested by Westphal et al. (2019). Some studies suggest that propionate and even butyrate may also be produced by the fermentation of organic matter (Sørensen et al., 1981; Lovley and Klug, 1986; Lovley and Phillips, 1989). The sporadic detection

of propionate in the samples could be indicative for this process. The degradation products can be used as substrate by other microbial communities. Cross-feeding interactions were described including sulfate reducing bacteria using lactate for sulfate reduction and in turn providing hydrogen for hydrogenotrophic bacteria in the Bad Blumau fluids, leading to the formation of acetate as a degradation product (Westphal et al., 2019). A gradual decrease of sulfate was only observed in the June 2021 samples, which might be related to the sulfate reduction in the power plant. However, it is more likely that these concentrations

represent its natural variability in the fluids since the high fluid flow provides continuous supply of sulfate.

Water-oil contact and an abiotic origin can likely be excluded due to the absence of other LMWA's, while the inhibitor only induces a minimal amount of acetate (2.2 µg C L$^{-1}$). To conclude, a microbial origin of the acetate due to fermentative processes is likely to account for the major part of the acetate present in our fluid samples.





## 5    Conclusions

With regard to the aims of this study, it was shown that 1) various methods to characterize the DOM were successfully used to give a detailed description of the DOM present in the fluids. It was shown with LC-OCD that the DOC consists predominantly of LMWAs and confirmed by IC that it is in fact acetate. Using FT-ICR-MS, the macromolecular DOM revealed to consist mainly of $O_x$ and $S_1O_x$ compounds, deriving mostly from the synthetic scaling inhibitor and meeting the aim 2) to distinguish between natural and synthetic OM. Overall, the scaling inhibitor adds approximately $1\,\mathrm{mg\,C\,L^{-1}}$ of artificial DOC to the fluids.

DOM compounds that were found in the inhibitor but not in the fluids such as $N_1O_x$ and $N_1S_1O_x$ might be absent due to (a) alteration in form of chemical reactions with the fluid forming complexes, (b) degradation after the injection into the fluids, (c) no detection in the fluid samples due to strong dilution of the inhibitor, (d) degradation by microorganisms that target these specific compounds. 3) The acetate is likely biogenic as a product of fermentative bacteria, turning lysine, pyruvate or hexitol into acetate. The dominant genera involved in these processes were *Bacillus* and *Desulfotomaculum*. 4) It was observed

that both the DOC concentration and the microbial communities change along the flow path. The change in DOC content is probably caused by microbial degradation. However, in this case, the temperature is likely the main driver of microbial community composition, since high temperatures are more limiting for microbial activity compared to the DOC concentration found here. However, in terms of DOM composition no significant change was observed along the flowpath and does not correlate with the changes in microbial diversity.

Additional insight in regard to chemical fluid composition, processes along the flowpath and origin of the organic matter in the Bad Blumau geothermal power plant was gained by assessing both, organic compounds and microbial composition in the fluids. These findings showed that both, the organic composition of a geothermal fluid and the microbial diversity are rather complex. So far no indications were found that they contribute to changes in productivity or injectivity of the wells. However, on a larger time scale the impact of the organics can still be relevant, e.g. when decomposition processes are not possible in

the inhibitor enriched fluids. Overall, this information adds significantly to the understanding of processes in the fluids of a geothermal site and might prove helpful to mitigate operational problems that could arise such as biofilm formation and/or microbially induced corrosion.

*Author contributions.*    AL analyzed the organic compound data. DB analyzed the microbial data. GC performed the modeling of the metabolic capabilities of the present microorganisms. AL prepared and wrote the major part the original draft with contributions from DB. AVH, FE,
SP, PJ, and SR reviewed and edited the manuscript. All authors read and approved the final manuscript.

*Competing interests.*    The authors declare that they have no conflict of interest.



*Acknowledgements.* We acknowledge the contribution of Kristin Günther and Cornelia Karger (both GFZ) for sample preparation and analyses of the organic compound, and The Spa Therme Blumau Betriebs GmbH for providing the fluid samples.



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
