# Peer review of "Methods to characterize type, relevance, and interactions of organic matter and microorganisms in fluids along the flow path of a geothermal facility"

_Biogeosciences, 2023_

## Author Response (AR1)

**Rebuttal**

Methods to characterize type, relevance, and interactions of organic matter and microorganisms in fluids along the flow path of a geothermal facility

The authors present a data report on the composition of fluids from a set of subsurface samples which have been mixed with a chemical additive. The compositional information is both chemical and biological assessments. I view this as a data report providing details about the water system, information which may be of interest to those who want to know what is in the water.

This manuscript will benefit from editing for grammar and language usage. There are also organization issues with the manuscript with results appearing for the first time in the discussion section, and details on methods also not appearing until the discussion. I am providing specific comments on the presentation in the sections that follow.

Thank you, for your comments. The reviewer addressed that this work is a data report on the composition of fluids from a set of subsurface samples. We disagree with this comment. This is not only a data report on subsurface water chemistry. This is a case study of a geothermal plant and the evaluation of the effects that this application and the usage of the subsurface will have on the chemistry and microbiology of these natural waters. We present and interpret our data and try to connect the results of the various methods that were used.

Thank you, for pointing out the organizational issues. We addressed them in your specific comments. We revised the manuscript for editing and language usage.

In the legend for figure 1 it says 'The black dots indicate the fluid sampling points…'; I see no black dots in the figure.

The legend text was corrected

Line 88: 'exsolution of CO2' – I don't know what exsolution means. Is there another translation for this word?

The sentence was rephrased and "exsolution" was substituted for "degassing". At that depth, the CO2 does not remain in solution anymore and is degassing.

The new sentence is at line 88: "The presence of CO2 in the system presents a major challenge to the operation of the plant. At a depth of approximately 300–350 m below the surface, CO2 degassing occurs, resulting in carbonate precipitation (Alt-Epping et al., 2013).

Line 92: 'polyelecrolytes' ? Please check this word as it is also unknown to me.

Was corrected to "polyelectrolytes"

Line 95  Are the units for CO2 really this: L L-1?

Yes

Line 107: I am unclear if six filters per site were processed, or six per day spread across the three sites.

We agree that this was not clearly explained and we modified the description as follows in the revised manuscript at line 105.

"For microbial analysis, 40 L of fluids were sampled at each of the three sampling points (P, HC, I) in June 2021. The 40 L of fluids were directly filtered through 0.22 µm nitrocellulose membrane filters (Merck Millipore, Germany) under sterile conditions. Filtering of each sample was done using an EZ-Stream pump (Merck Millipore, Germany) and six glass filtration stations mounted on a manifold (Merck Millipore, Germany). This resulted in six filters for each sample that were prepared simultaneously and served as independent replicates. Filters were transported to the laboratory at a temperature of 4 °C and were later stored at -20 °C until further processing."

Line 109: -20C seems a little warm to store filters for DNA analysis. How long were the filters stored at this temperature, and do the authors have evidence that there was no degradation of the DNA signal at this temperature?

The filters were stored at -20°C to avoid introducing bias by sequencing all the samples generated within the project in different sequencing batches. This would have impaired our ability to do comparisons within different geothermal plants as for samples with such a low biomass, the impact of intra sequencing batch variation would have been expected to be very large. Unfortunately, that meant that the samples were stored for 16 months due to a delay in laboratory analyses related to the Covid-19 pandemic. We have performed extensive research of the literature available concerning the impact of storage on microbial communities. Unfortunately, these types of studies have been mostly performed on sediments or on fecal samples (human gut microbiome research). Those studies (references below) suggest that even in the presence of the sample matrix, storage at even room temperature or even drying, results in a reproducible representation of the community. We could not find any study systematically assessing the impact of storage at -20°C in filtered water communities suggesting that the temperature of storage would have caused a bias that affected differentially one or the other samples in comparison in this study. Moreover, it would be worth highlighting that the community composition obtained is in agreement with previous studies in the same site, suggesting that the method of storage should not have resulted in a dramatic change in the community, affecting the interpretation of the data.

References assessed:

Investigating the Impact of Storage Conditions on Microbial Community Composition in Soil Samples

https://journals.plos.org/plosone/article?id=10.1371/journal.pone.0070460#s2

Effect of storage conditions on the assessment of bacterial community structure in soil and human-associated samples

https://academic.oup.com/femsle/article/307/1/80/472147

Characterization of the Fecal Microbiota Using High-Throughput Sequencing Reveals a Stable Microbial Community during Storage

https://journals.plos.org/plosone/article?id=10.1371/journal.pone.0046953

Drying as an effective method to store soil samples for DNA-based microbial community analyses: a comparative study

https://www.nature.com/articles/s41598-023-50541-2

Line 119: I am unclear on how you use Cl as a means to determine if the chloride concentrations are too high. Do you have an external means to measure Cl before the IC measurement? And, from the later results section it would appear that all the samples had Cl measurements that would require this pre-processing step.

Very high chloride concentrations result in enormous peaks in the IC-chromatogram disturbing the small peaks of the organic acid anions eluting directly behind the chloride. This is the reason why we need an additional pre-processing step for samples with high chloride concentrations. For the Bad Blumau site, we already knew that the chloride concentrations in our fluids would be >1 g/L based on literature data and information from the site operators and this pre-processing was applied for Bad Blumau samples prior to organic acid analysis.

For other samples with unknown chloride concentrations, a first test measurement will be done by IC and based on the chloride concentration measured here, the additional pre-processing step will be applied before the IC analysis for the organic acids. There is no external means or a different analytical method for this first check and estimation about the chloride concentration.

Line 139: the methanol and acidified water are used to activate the resins for use and are not a cleaning step.

Thanks for the correct statement, we changed it in the revised manuscript (line 142).

Line 225: 'The DOC in the fluid samples ranges from 8.4–10.3 mg CL-1 (Table 2), showing a decrease along the pathway. In March, the DOC decreased from the production to the injection side whereas in June the concentrations seemed relatively

uniform with slightly higher DOC in the heat central.' – this is confusing because the first broad statement actually only applies in March.

The statement clarified by deleting "showing a decrease along the pathway". Here is the new statement in the revised manuscript (line 227): "The DOC in the fluid samples ranges from 8.4 – 10.3 mg CL-1 (Table 2). In March, the DOC decreased from the production to the injection side whereas in June, the concentrations are quite similar in production and injection sides and slightly higher at the heat central.

Comment from the authors: Bad Blumau geothermal plant is monitored very regularly for basic chemical parameters since many years. And here, there is the overall trend of decreasing DOC concentrations along the pathway. Unfortunately, this trend is not that clear for the two sampling campaigns that we investigated in many more detail here. But we will compare the data of the two campaigns we did with the long-term monitoring data in the discussion.

Line 252: "The fluid samples are affected by inhibitor signals ranging from 31–65.3 %TMIA." I am unclear on what this means – how can the fluid samples be 'affected' by the inhibitor? In the table also lists an SI amount as a percent TMIA, but I see no description of how this was defined. Perhaps this was the description starting at line 315 in the discussion, and if so that definition needs to be moved to the method section. And, if that is the description, I am still unclear on how the compounds were added – is this a count of the number of compounds in the inhibitor or is this a summing of the peak intensities? Are these the compounds that are unique to the inhibitor, or is this any compound in the inhibitor?

1. The term "affected" has been removed and the sentence rewritten as follows: "In direct comparison, 31 – 65.3 % of the TMIA in the fluid samples derive from formulas that are also present in the inhibitor samples.". Revised manuscript (line 254).
2. In Table 3, the column named "SI amount" gives the relative proportion of TMIA in the fluid samples that derives from compounds that are also detected in the inhibitor. E.g. SI amount of 65.3 %TMIA means that 65.3 %TMIA in this sample derive from compounds that were also detected in the inhibitor sample and might have been introduced by the inhibitor to the fluid. It is a means to show how much of the complex organic matter in the fluid samples might derive from the inhibitor.

3. The description on Line 315 will be moved to the method section. Additionally, we will clarify that this value is the summing of peak intensities that are unique to the inhibitor (Line 178).

Table 2 – presents information with an excessive number of significant digits. Sokal and Rolf provide a nice guideline about keeping significant digits between 30 and 300 unit steps (a quick Google search provides the full description).

This advice was applied for table 2 and in the text.

Line 292: 'However, it is important to note that the top five genera at each sampling point does not represent the entire community, but only a subfraction of the most abundant members of the community.' I am not sure why this is important to note, this seems to just be repeating the definition of the top five genera and is not a notable scientific result.

We agree, this sentence was removed

Figure 5: What date are these diversity samples from? The methods describe two dates for the chemical data, but is not clear as to when the bacterial diversity samples were obtained.

The samples come from the June 2021 sampling campaign. The specific dates were included in the methods. Now in the revised manuscript (line 105).

Figure 6: I am unclear what the x-tick labels are here. #s 55 to 69 are presumably sample sites, but this has not been defined. Also, what is the y-axis? Is 'contrib' the number of strains involved in each process? Or the relative magnitude of the amount of chemical that would be altered by this process?

The figure and the legend of the figure will be modified to improve clarity. The "Contribution" term will be replaced by "relative abundance". The legend text will be changed to:

Top 5 Genus per sample and their relative abundance in (a) lysine to acetate and butyrate fermentation, (b) pyruvate fermentation to acetate and lactate II, (c) hexitol fermentation to lactate, formate, ethanol and acetate, (d) methanogenesis from acetate. Sampleid (55 – 69) represents the numbers assigned to each filter.

Line 310: 'identify the inhibitor-derived organic compounds' – you did not identify the compounds as that would require additional analyses with authentic standards. I would use 'characterize' as a more generic phrase.

Thanks for this hint. We changed it to 'analyze'. Revised manuscript (line 320).

Line 311: 'is the distinction and quantification between synthetic and natural organic matter possible?' – this is the authors' first question and cannot be addressed given the samples in hand. The study lacks the controls that would be required to answer this question as there is sample without the synthetic material. It is also unclear to me as to why you would want to address this question – the inhibitor will always be in these samples.

We agree and the sentence has been rephrased to: "is it possible to characterize and subtract synthetic organic matter to obtain an indication of the natural organic matter in geothermal fluids?" (revised manuscript, line 321).

We are completely aware that there is no water sample available without any inhibitor. But we have a pure sample of the inhibitor that will be added to the fluid. And we therefore decided to compare the composition of the fluid with the composition of the inhibitor and to "subtract" the inhibitor signals from the fluid signals to get an indication about the natural organic matter in the fluid. And as also stated in the manuscript – this is the first time that such a sample (fluid from a very deep reservoir that is used for geothermal energy production) is investigated by FT-ICR-MS.

The first 1.5 pages of the discussion are results, which should not be appearing for the first time in the discussion section of this manuscript.

Yes, this part still presents our data but we also start to discuss in detail how much inhibitor can be found in the fluids along the flow path. We compare inhibitor and fluid DOM and start to make assumptions which compound classes derive from the inhibitor or why some inhibitor compounds cannot be found in the fluids (to name a few). However, we will revise this part. Data that appears here for the first time will be moved to the result section.

As a result of the restructuring, figure 3 and 7 were swapped.

Line 345: 'differ clearly' – this is vague. Please provide the statistical tests needed to support this ascertation.

There is a significant difference between the mean oxygen number of the fluid and inhibitor samples (p-adj. = 0.008**). However, we are not focusing on statistical analyses in this manuscript. We rephrased the sentence to:

"The mean oxygen numbers decrease from the production to the injection side (10 to 9.52) and are lower than the mean oxygen number of the inhibitor (12.8).". Revised manuscript (line 278).

Line 420: 'Thus, slight variations and differences such as inhibitor signals being present in the heat central and not in the other samples do not necessarily represent processes linked to the flowpath.' This statement by the authors is odd since in the other portions of the discussion they specifically link changes in the fluid composition to the flow path. Why are they now saying that these fluids cannot be linked to the flowpath?

We agree, this is contradictory. We removed this statement. However, we clarified that the FT-ICR-MS data is not quantitative and therefore we were not able to see if there were changes along the flow path. Following statement was made in the revised manuscript (line 418): "It is important to note that FT-ICR-MS is not a quantitative

approach and therefore no assumptions to concentrations along the flow path can be made with these data."

Line 508: '…make it less likely that the organic acid anion content is associated with the water-oil contact.' This ignores the possibility that microorganisms have already consumed those anions before the water sample was collected.

Indeed, this is true. We included this possibility with this sentence in the revised manuscript (line 506): "However, the consumption by microorganisms could be another reason for the absence of these organic acids in the fluid samples."

Given that the biochemical predictions were used to assess the presence of acetate, why did the authors not also consider the biochemical pathways that would lead to the absences of the other compounds that could have been measured with the IC system?

We only checked for acetate pathways because this was the anion that was clearly detected in our samples. We decided to focus on this aspect.

Review of "Methods to characterize type, relevance, and interactions of organic matter and microorganisms in fluids along the flow path of a geothermal facility" by Leins et al.

In this manuscript, the authors analyzed the dissolved organic matter and microorganisms along the flow path of a geothermal facility. They used FT-ICR MS to characterize and differentiate between natural and synthetic organic matter, and determine the implications of microorganisms in an operating a geothermal site. Overall, there is a good deal of data being analyzed and the samples appear interesting. However, there are substantial gaps in the explanation for the methods as well as some of the interpretations that must be addressed before this work is published. I consider these major revisions and these are listed in the specific comments below.

Line 11: Please explain what type of compounds are "macromolecular formulas".

Explanation was added (Line 10). We mean organic compounds with a molecular mass of 150 Da to 1000 Da.

Line 50: The common ionization source used to analyze DOM in general is the electrospray ionization. Please describe the need to use the APPI in the introduction.

We used APPI because the chemical properties of the inhibitor compounds were not known before. APPI is supplementary or even complementary to ESI since it is able to ionize less polar compounds.

This sentence was included in the revised manuscript at line 50: "Both APPI(+) and ESI(-) ionization modes were used to detect polar and less polar compounds since the chemical properties of the inhibitor were unknown beforehand"

Line 150: The addition of ammonia is generally not required when analyzing DOM. Please explain why ammonia was used in this study.

We are using NH4OH for all ESI (neg) samples that we analyze. While it might not be necessary for DOM, we wanted to be sure in case that inhibitor compounds would need "assistance" for their ionization.

Line 158: Why is toluene not added when analyzing the DOM using APPI? How do you achieve ionization using only methanol?

We use here for the measurement solution a mixture of n-hexane and methanol, and not purely methanol. Unfortunately, this one sentence concerning the measurement solutions went somehow missing for the APPI part and we are glad that the reviewer mentioned this. For APPI, you need at least one solvent with an IE that is below the energy that is provided by the laser (Krypton Laser with 10.6 eV). In our case this is n-hexane with an IE of 10.13 (<10.6). With n-hexane instead of toluene, you produce almost exclusively protonated ions and no radical ions in complex mixtures, this is a great advantage because it reduces the complexity of the samples.

The missing sentence was included in the revised manuscript at line 160: "For the APPI(+) analyses, solutions of 20 µg ml-1 in MeOH were prepared from the stock solutions. Measurement solutions of 100 µgml-1 were prepared in MeOH:n-hexane (9:1)."

Line 174: When assigning molecular formulas, why do you set the number of N and S elements to 1? Are there no molecular formulas containing 2 N atoms or 2 sulfur atoms in any of the samples in this study?

We indeed checked for 2 N and two S atoms within our search, but found only compounds with one N and one S which is not surprising for DOM. We will correct the elemental thresholds for N and S from 1 to 2 in the methods part.

Line 187: Whose work is referenced for the classification of compounds here?

Koch and Dittmar, 2006 for >0.5 (aromatic) and >0.67 (condensed aromatic). This reference is already mentioned in the sentence and Zhu et al., 2019 for <0.5 (aliphatic).

The revised manuscript now contains following sentence at line 191: "Three ranges were established to describe the aromaticity of a given DOM compound. AImod values ≤ 0.5 are described as aliphatic (Zhu et al., 2019), AImod between 0.5 and 0.67 represent aromatic compounds and AImod ≥ 0.67 describes condensed aromatic compounds (Koch and Dittmar, 2006)."

Line 241: What samples were actually analyzed using FT-ICR MS in this study? Here it says "LMWA and Makro fractions in ESI(-) and from LMWN and Makro fractions in APPI(+) mode", but the figure and table show the results from the original collected water samples.

1. The March 2021 samples were analyzed using FT-ICR-MS. This detail was added in the revised manuscript at line 140.
2. The above mentioned sentence is meant to describe which range of the DOM that was uncovered by LC-OCD analyses can be analyzed in more detail by FT-ICR-MS. E.g. APPI(+) and ESI(-) FT-ICR-MS enables 150 Da to 1000 Da of molecular mass range. The DOC fractions detected by the LC-

OCD analyses are split into Makro (350 Da to 10 000 Da), LMWA (< 350 Da) and LMWN (< 350 Da) fractions. This is also shown in Table 1. So, the overlapping range of both methods lies at 150 Da to 1000 Da.

The sentence was changed in the revised manuscript (line 244): "While LC-OCD analyses provides general information about the molecular size distribution of the DOM, FT-ICR-MS enables highly-resolved insight into the molecular composition of DOM compounds within a mass range from approximately 150 to 1000 Da. Acidic compounds are detected in ESI(-), and low polarity neutral compounds are characterized using in APPI(+) mode."

Line 259-263: This discussion is not a result of this study

The paragraph was moved to the discussion in the revised manuscript at line 337.

Section 3.3: The authors have used a great deal of space in this section to describe the results obtained from the two ionization sources. Theoretically, different molecular formulas can certainly be arrived at using different ionization sources. But how do the results fit between the two ionization sources? Is it a complementary relationship? Judging from the results so far, it should not be. In addition, is it possible to do a semi-quantitative analysis between the results obtained from the two ionization sources?

Of course, ESI and APPI are able to ionize different types of compounds. The main difference is the degree of polarity within the molecules. While ESI(-) mainly ionizes acidic compounds, APPI(+) is able to ionize a broader range of aromatic compounds with non-acidic functionalities. Therefore, we say that both are complementary. However, this complementarity cannot be simply seen from the elemental formulas, since compounds with the same elemental formula detected by the different modes are different compounds. Mention water solubility as overall framework for type of compounds. We did not aim for semi-quantitative analysis.

Line 317: Were internal standards added for semi-quantitative analysis?

No, there was no internal standard added. We did not aim for a semiquantitative analysis.

Line 372: How comparable is the DOM in Sample "dismal swamp and pore water of a Mangrove" to the DOM covered in this study? What is the purpose of mentioning these 2 references here?

We agree here with the reviewer. We changed the citations and compare now our geothermal plant DOM samples with groundwater samples (McDonough et al., nature communications 2022, https://doi.org/10.1038/s41467-022-29711-9) at line 369 in the revised manuscript.

Section 4.1: Typical natural DOM samples contain several thousand molecules, but only a few hundred in this study. Is it comparable to compare the samples in this thesis with DOM such as seawater?

This is a result of concentration differences. In our samples, the inhibitor compounds are much more abundant then the DOM compounds. Therefore, since all compounds compete for ionization, DOM is ionized to a lower degree compared to natural DOM samples and of course the least abundant constituents are now below the signal to noise ratio of the measurements.

Line 409: Why compare the DOM in this study to the ocean DOM? The two do not have similar geologic environments.

We removed the seawater sample but decided to keep the hydrothermal vent samples since hydrothermal and geothermal settings are comparable. This part is now at line 406 in the revised manuscript.

Section 5: The conclusions of this paper and need to be condensed again.

The conclusion was condensed.

---

## Author Response (AR2)

Dear Mr. Leins and co-authors:

First my apologies for the very slow processing of your revised paper. I have tried my best to obtain reviews but failed. This is exceptional and very unfortunate. It did never happen before in more than 50 years of cumulative associate editorships. One of the original referees was not interested to see it again, and the other did not respond, despite multiple system and three personal reminders. Other referees contacted also declined. I have therefore decided to have a more detailed look myself rather than further contacting referees.

Your revised paper has very much improved, the data are of high quality and quite interesting, but there is scope for further improvement.

Answer:

Dear Dr. Middleburg,

Thank you for taking the time to review the manuscript and for your feedback. We have addressed your suggestions and made the necessary adjustments to the manuscript.

**1)** Although the organization of the paper is better than before, I suggest re-organizing the results section so that the results referring to the inhibitor only and to the sample water (that contains the inhibitor) are better separated. This will improve the readability much. The reader is still somewhat lost in the results section.

**Answer:**

The main organization of the Results follows the methodology that was applied to the samples (Anion analyses, DOC and bulk fraction, FT-ICR-MS analyses). In our opinion it is not favorable to split the results into fluid and inhibitor sample for the first two since the data for the inhibitor in these subsections is minimal. However, we followed your suggestion for "3.3 Molecular composition of the DOM" and hope that this improves the readability. Due to the new structure we incorporated the mean DBE and $AI_{mod}$ values in Table 3.

**2)** The abstract needs a major rewrite.

a. Write more active: For instance, the first sentence could Dissolved organic matter (DOM) and micro-organisms were characterized along…. The next sentence then Various analytical methods were used to differentiate…

b. The flow and logic: 'the inhibitor' suddenly appear in line 7 without prior explanation what type of inhibitor and why etc. This comes a few lines later.

c. Prevent abbreviation as much as possible. For instance, a sentence like using PICRUSt2 is too cryptic. Write …using amplicon sequence variants. (the precise pipeline used is not needed in the abstract).

**Previous Abstract**

[revised manuscript text omitted]

3) The introduction might sometimes give the impression that the chemical and molecular biology analyses are more important than the research question (this was communicated by the referees before). For line 42 states that IC was applied for information on… Wouldn't it be more interesting to present the problem that an inhibitor has been added and that its composition is not known because of commercial interest and that you have therefore analyzed the inhibitor and the fluids so that it can trace the relative contribution and transformation of the inhibitor.
**Answer:**
Part of the introduction was rewritten (start at line 43). It now presents the problem of having an inhibitor with unknown composition in the fluids and which techniques were applied to trace inhibitor and fluid DOM.

4) The use of Generally, in line 92 is unclear. Turn into However, ….
**Answer:** Done

5) The unit 5 L $L^{-1}$ needs more explanation: L $L^{-1}$ units are sometimes used for gases but then it should be less than one. Does the one volume refer to another phase than the other volume? This was identified by one of the referees before.
**Answer:**

This sentence was clarified (Line 102): The $CO_2$ concentration makes up 99 % of the gases and is estimated to be approximately 10 g L-1, which correlates to approximately 5 L of $CO_2$ per L water.

**6)** The use of its in line 68 is unclear. Does it refer to natural and synthetic DOM?
**Answer:**
We restructured the sentence for clarity (Line 75): (2) determine the origin of DOM by distinguishing between natural and synthetic sources.

**7)** L. 85: at approximately
**Answer:** Corrected

**8)** Line 123: size exclusion chromatography in full rather than directly using SEC.
**Answer:** Corrected

**9)** Line 329: a strong influence of what?
**Answer:**
Rephrased the sentence for clarity (Line 360): The assigned signals from the fluid and inhibitor samples indicate a strong influence of the inhibitor on the production side sample, with contributions reaching up to 65 %.

---

## Author Response (AR3)

**Author response**

Dear Dr. Middleburg,

Thank you for your fast reply and accepting the final version of the manuscript. All requested corrections were made.

with best regards,

Alessio Leins

**Corrections:**

Line 7: delete second characterizing

Corrected

Line 107: CO2 subscript
Corrected

Line 135: .. the DOC was separated..
Corrected

Line 151: pH 2

Corrected

Line 184: ..2, and unlimited C and H. The mass…

Corrected

Line 450: … and therefore no conclusions/inferences regarding ….

Corrected

Line 461: do you mean site or side? Same in line 462.

Side is correct. When referring to the sampling point, we use "side" as in production side or injection side of the geothermal facility. An example of when the term "site" is used is "geothermal site" or "site operator". We have checked for consistency and it is now consistent throughout the manuscript.

Line 493: …2019). In that study, samples…

Corrected

Line 498-499: check consistently in use of italics (all through).

We have corrected and checked the manuscript for consistency. By convention, family, genus and species names should now all be in italics, while kingdom, phylum, class, order and suborder levels remain normal.